# LSEWOA: An Enhanced Whale Optimization Algorithm with Multi-Strategy for Numerical and Engineering Design Optimization Problems

**DOI:** 10.3390/s25072054

**Published:** 2025-03-25

**Authors:** Junhao Wei, Yanzhao Gu, Yuzheng Yan, Zikun Li, Baili Lu, Shirou Pan, Ngai Cheong

**Affiliations:** 1Faculty of Applied Sciences, Macao Polytechnic University, Macao 999078, China; p2312195@mpu.edu.mo (J.W.); p2311998@mpu.edu.mo (Y.G.); p2312012@mpu.edu.mo (Y.Y.); 2School of Economics and Management, South China Normal University, Guangzhou 510006, China; 18520610821@163.com; 3College of Animal Science and Technology, Zhongkai University of Agriculture and Engineering, Guangzhou 510225, China; 18023304003@163.com (B.L.); 13798144163@163.com (S.P.)

**Keywords:** WOA, Spiral flight, Tangent flight, engineering design, inertia weight, numerical optimization

## Abstract

The Whale Optimization Algorithm (WOA) is a bio-inspired metaheuristic algorithm known for its simple structure and ease of implementation. However, WOA suffers from issues such as premature convergence, low population diversity in the later stages of iteration, slow convergence rate, low convergence accuracy, and an imbalance between exploration and exploitation. In this paper, we proposed an enhanced whale optimization algorithm with multi-strategy (LSEWOA). LSEWOA employs Good Nodes Set Initialization to generate uniformly distributed whale individuals, a newly designed Leader-Followers Search-for-Prey Strategy, a Spiral-based Encircling Prey strategy inspired by the concept of Spiral flight, and an Enhanced Spiral Updating Strategy. Additionally, we redesigned the update mechanism for convergence factor *a* to better balance exploration and exploitation. The effectiveness of the proposed LSEWOA was evaluated using CEC2005, and the impact of each improvement strategy was analyzed. We also performed a quantitative analysis of LSEWOA and compare it with other state-of-the-art metaheuristic algorithms in 30/50/100 dimensions. Finally, we applied LSEWOA to nine engineering design optimization problems to verify its capability in solving real-world optimization challenges. Experimental results demonstrate that LSEWOA outperformed better than other algorithms and successfully addressed the shortcomings of the classic WOA.

## 1. Introduction

Metaheuristic algorithms are a class of algorithms that combine random algorithms and local search algorithms, aiming to solve a variety of complex optimization problems. In recent decades, metaheuristic algorithms have been developed, extensively studied, and widely applied. Due to the complexity and diversity of many real-world problems, canonical exact algorithms often struggle to find optimal solutions within a reasonable time frame. Metaheuristic algorithms are developed based on heuristic algorithms, and their core idea is to progressively approximate the optimal solution by searching through the problem space. They can provide feasible solutions within acceptable computational time and space, although the degree of deviation from the optimal solution may not be predictable in advance. The main advantage of metaheuristic algorithms is their ability to handle complex, nonlinear problems without requiring assumptions about the specific models of different problems. Metaheuristic algorithms primarily consist of two phases: a global exploration phase and a local exploitation phase. Since the optimal solution may exist at any location within the search space, the exploration phase aims to explore the entire search space as thoroughly as possible. The exploitation phase focuses on utilizing effective information, as in most cases, there is often some correlation between superior solutions, and these correlations are used to progressively adjust and find better solutions. Compared to other numerical optimizers, metaheuristic algorithms combine both global and local search strategies. By integrating the exploration and exploitation strategies and controlling the balance between them, they increase the probability of finding the global optimal solution in complex problems. Additionally, metaheuristic algorithms introduce randomness into the search process to avoid being trapped in local optima, thereby enhancing their ability to find better solutions. Metaheuristic algorithms do not require problem-specific knowledge and typically do not rely on the specific nature or structure of the problem, making them highly versatile and capable of handling a variety of optimization problems. They also exhibit a degree of randomness to prevent premature convergence to local optima. Moreover, metaheuristic algorithms are adaptive, capable of adjusting parameters and strategies in real time based on the progress of the optimization process.

Given the many advantages of metaheuristic algorithms in numerical optimization, they have received increasing attention, with more and more researchers developing improved metaheuristic algorithms to better solve complex optimization problems. Some classic optimization algorithms are shown in Table 1 and Table 2. The Simulated Annealing (SA) algorithm, proposed by Metropolis in 1953, is based on comparing the process of solving a certain class of optimization problems to the thermal equilibrium problem in statistical thermodynamics, attempting to find the global optimal solution or an approximate optimal solution by simulating the annealing process of high-temperature materials [1]. The Genetic Algorithm (GA), first introduced by John Holland in 1975, is based on Darwin’s theory of evolution and Mendel’s genetics. It uses processes such as reproduction, mutation, and competition among individuals in a population to exchange information and perform natural selection, progressively approaching the optimal solution to the problem [2]. The Ant Colony Optimization (ACO) algorithm, proposed by Dorigo et al. in 1991, is a stochastic search algorithm that simulates the foraging behavior of real ants in nature [3]. In 1995, American psychologists Kennedy and electrical engineer Eberhart, inspired by the foraging behavior of birds, proposed the Particle Swarm Optimization (PSO) algorithm [4]. Storn and Price, in 1997, proposed the Differential Evolution (DE) algorithm for real-valued function optimization [5]. In 2005, Karaboga et al., inspired by the honeybee foraging behavior, proposed the Artificial Bee Colony (ABC) algorithm, which simulates the intelligent foraging behavior of bee colonies and compares the global optimal solution to the nectar-richest flower source in the search optimization problem [6]. Elhamifar and Ahmadi, inspired by the hunting behavior of Harris hawks, proposed the Harris Hawks Optimization (HHO) algorithm in 2019 [7]. In 2021, Laith Abualigah et al. proposed the Aquila Optimizer (AO), which was inspired by the hunting behavior of aquila eagles in nature [8]. In 2022, C Zhong et al., inspired by the swimming, foraging, and whale fall phenomena of beluga white whales in nature, proposed the Beluga whale optimization algorithm (BWO) [9]. In 2023, Jia et al., inspired by the social behaviors (foraging, cooling, and competitive behaviors) of crayfish in nature, proposed the Crayfish Optimization Algorithm (COA) [10]. Metaheuristic algorithms, due to their excellent optimization capabilities and versatility, are widely applied in various fields, such as Advanced Planning and Scheduling (APS) [11], engineering design optimization [12], feature selection [13], urban site selection [14], path planning [15], the traveling salesman problem [16] and antenna design optimization [17].

However, metaheuristic algorithms also have certain limitations. These algorithms often face challenges such as the difficulty in balancing exploration and exploitation, difficulties in parameter selection, poor population quality in the later stages of iteration, and the tendency to become trapped in local optima. Particle Swarm Optimization (PSO) faces the risk of premature convergence when dealing with complex optimization problems, often leading to early convergence to local optima, halting further exploration of better solutions. The Harris Hawks Optimization (HHO) algorithm is known for its strong exploration and exploitation capabilities, but its numerous parameters and complex position update strategies make its tuning more difficult. The Whale Optimization Algorithm (WOA) is simple in structure and easy to implement, but it suffers from slow convergence, low solution accuracy, and the tendency to get trapped in local optima in complex problems [18]. Therefore, improving the balance between exploration and exploitation, enhancing the efficiency of the exploration phase, increasing the accuracy of the exploitation phase, and maintaining population diversity in the later stages of iteration have become the major challenges in enhancing the performance of metaheuristic algorithms.

**Table 2 sensors-25-02054-t002:** Current research on improved metaheuristic algorithms.

Algorithm	Year	Author	Source of Inspiration
COGWO2D [19]	2018	Ibrahim R A et al.	Opposition-Based Learning, Differential Evolution, and disruption operator.
MEGWO [20]	2019	Tu Q et al.	Adaptable cooperative strategy and disperse foraging strategy.
QMPA [21]	2021	Abd Elaziz M et al.	Schrodinger wave function.
ISSA [22]	2023	Xue Z et al.	Circle chaotic mapping, GWO and chaotic sine cosine mechanism
ACRIME [23]	2024	Abdel-Salam M et al.	Symbiotic Organism Search (SOS) and restart strategy
BWOA [24]	2019	H Chen et al.	Levy flight and chaotic local search.
SMWOA [25]	2020	W Guo et al.	Linear incremental probability, social learning principle and Morlet wavelet mutation
HSWOA [26]	2021	VKRA Kalanandan et al.	Social Group Optimization algorithm (SGO)
ImWOA [27]	2022	S Chakraborty et al.	Cooperative hunting strategy and improving the exploration-exploitation logic

In recent years, researchers have continuously attempted to integrate different methods to improve metaheuristic algorithms. In 2004, Y. Gao et al. introduced chaotic mapping into population initialization, generating higher-quality populations, which enhanced the optimization ability of the Particle Swarm Optimization (PSO) algorithm to some extent. In 2018, Rehab Ali Ibrahim et al. proposed the Chaotic Opposition-based Grey Wolf Optimizer (GWO) based on Differential Evolution and Disruption Operator. They incorporated logistic mapping, opposition-based learning (OBL), differential evolution (DE), and disruption operator (DO) into GWO to improve its exploration and exploitation capabilities while maintaining population diversity [19]. In 2019, Qiang Tu et al. introduced the Multi-strategy Ensemble Grey Wolf Optimizer (MEGWO), which incorporated the Enhanced Global-best Leading Strategy, Adaptable Cooperative Hunting Strategy, and Scattered Hunting Strategy into the canonical GWO to overcome the limitations of a single search strategy when solving various optimization problems [20]. In 2021, Mohamed Abd Elaziz et al. proposed the Quantum Marine Predators Algorithm (QMPA), which uses the probability function from the Schrödinger wave function to determine the position of particles at any given moment, thus enhancing the exploration and exploitation abilities of the Marine Predators Algorithm (MPA) [21]. In 2023, Zhilu Xue et al. proposed the ISSA, which introduced chaotic mapping, integrated the information exchange mechanism from the Grey Wolf Optimizer (GWO), and utilized chaotic sine-cosine strategies to improve the optimization accuracy and convergence speed of the Social Spider Algorithm (SSA) [22]. In 2024, Mahmoud Abdel-Salam et al. introduced chaotic mapping, adaptive improvement of the Symbiotic Organisms Search (SOS) mutualistic phase, hybrid mutation strategies, and restart strategies into the RIME algorithm, proposing the Chaotic RIME Optimization Algorithm with Adaptive Mutualism (ACRIME) to address feature selection problems [23]. These outstanding algorithms, which integrate various novel improvement strategies, provide new insights for enhancing metaheuristic algorithms.

The Whale Optimization Algorithm (WOA), proposed by Mirjalili et al. in 2016, is a metaheuristic optimization algorithm inspired by the hunting behavior of humpback whales [18]. WOA has a relatively simple structure, making it easy to understand and implement. However, WOA suffers from poor balance between exploration and exploitation, and the population quality tends to degrade significantly during iterations, leading to insufficient global exploration and premature convergence to local optima. These drawbacks make WOA less competitive when solving complex and real-world optimization problems. As a result, many researchers have attempted to improve the performance of WOA in recent years. In 2019, Huiling Chen et al. proposed an improved Whale Optimization Algorithm (BWOA), which integrates Levy flight and chaotic local search strategies (CLS) to more effectively balance global exploration and local search capabilities [24]. In 2020, Guo W et al. introduced a modified Whale Optimization Algorithm (SMWOA), which incorporates social learning and wavelet mutation strategies. This algorithm significantly enhances global search efficiency by designing a new linearly increasing probability [25]. In 2021, Vamsi Krishna Reddy Aala Kalananda et al. proposed two new hybrid optimization algorithms, named Hybrid Social Whale Optimization Algorithm (HS-WOA and HS-WOA+), which combine the advantages of WOA and Social Group Optimization [26]. These algorithms integrate WOA’s exploration ability with SGO’s convergence properties, achieving a perfect balance between exploration and exploitation. In 2022, S Chakraborty et al. introduced a novel improved Whale Optimization Algorithm (ImWOA), which employs two different exploration strategies to explore food sources and introduces a new collaborative hunting strategy [27]. This was designed to address the issues of solution diversity and local optima in the canonical WOA. However, although these algorithms achieve a good balance between exploration and exploitation, they fail to effectively improve convergence speed and accuracy. Therefore, this paper proposed an enhanced Whale Optimization Algorithm based on multi-strategy (LSEWOA) to address these issues.

## 2. Development History and Current Research of Engineering Design

In ancient times, the design of structures largely relied on experience and intuition. When constructing the pyramids in ancient Egypt, designers determined the dimensions and shapes of the structures based on human experience and mathematical principles. In ancient architecture or mechanical design, the concept of “optimization” was generally absent, with designs being based more on manual calculations and intuitive judgment to create reasonable structures. During the Industrial Revolution, with the advancement of machine manufacturing and mass production, engineering design began to focus on optimization to enhance structural performance and reduce costs, although it still relied on manual design. Optimization was typically achieved through experimentation, adjustment, and correction. In fields like thermodynamics and mechanical structural design, designers would conduct elaborate experiments to adjust design parameters until they met practical requirements.

In the early 20th century, mathematical optimization theory gradually developed and began to serve as a core tool in engineering design. The initial optimization methods were based on classical mathematical analysis, such as calculus, which provided optimal solutions through analytical derivations. In the 1940s, linear programming (LP) was proposed by George Dantzig and others, providing a mathematical foundation for optimization, particularly in applications such as economics, transportation, and resource allocation. With the emergence of nonlinear systems, designers required more complex mathematical tools to solve these problems. As a result, numerical methods like Newton’s method and gradient descent were developed and applied to engineering optimization, addressing more complex design problems. However, due to the lack of powerful computational tools at the time, engineering design remained a time-consuming and labor-intensive process. The optimization process required designers to make certain assumptions to find solutions, and the problems solved were often simplified or idealized.

From the 1950s to the 1970s, with the development of computers, engineering design optimization gradually became computerized. Engineers began using computer programs to solve optimization problems, and breakthroughs were made, particularly in finite element analysis (FEA) and dynamic programming (DP). The finite element method allowed engineers to model and analyze complex structures, solving stress and deformation problems for various materials and geometries, providing more detailed and accurate computational tools for engineering optimization. In control theory and scheduling problems, dynamic programming methods were widely applied, breaking down problems into smaller subproblems and optimizing them step by step, solving many complex design issues. However, in these early computer-aided designs, optimization problems were typically solved using deterministic mathematical methods and were limited to certain types of problems (e.g., linear and nonlinear issues). When faced with large-scale, complex engineering problems, the computational effort and difficulty in solving these problems remained substantial.

As the scale and complexity of engineering design problems continued to increase, canonical mathematical optimization methods struggled to handle high-dimensional, nonlinear, and complexly constrained problems. By the 1980s, researchers began to turn to metaheuristic algorithms to address complex engineering optimization problems. Metaheuristic algorithms, such as simulated annealing, genetic algorithms, particle swarm optimization, and ant colony optimization, were developed and put into practical use. These algorithms effectively avoided getting trapped in local optima and provided better optimization solutions, offering a new approach to engineering design optimization, especially for complex, irregular, or high-dimensional problems. At the same time, significant improvements in computer hardware enabled these algorithms to handle larger-scale optimization problems. In the 2010s, the rapid development of artificial intelligence (AI) and deep learning (DL) began to offer additional options for engineering design optimization. For example, neural networks (NN) can automatically predict and help designers optimize structural designs by learning from large amounts of historical data. However, AI models require extensive training time and large datasets to perform effectively, and for engineering design problems where data is scarce or difficult to obtain, their performance may be suboptimal. Additionally, AI depends on the mathematical model of the problem, and once trained, it can only handle known engineering design problems within that model.

Given that metaheuristic algorithms are robust and maintain high optimization performance across different application scenarios, typically without relying on the mathematical model or derivative information of the problem, they are well-suited for various complex and difficult-to-model problems. Consequently, metaheuristic algorithms remain the primary method for solving engineering design optimization problems today.

## 3. Organization of the Paper

Section 4 primarily presents the main contributions of this research. Section 5 provides a detailed explanation of the principles of the WOA, along with its advantages and disadvantages. Section 6 introduces the proposed LSEWOA proposed in this paper. In Section 7, we will evaluate the performance of LSEWOA through a series of experiments. Section 8 involves testing various metaheuristic algorithms and LSEWOA on different engineering design optimization problems to validate the practicality and robustness of LSEWOA.

## 4. Major Contributions

Whale Optimization Algorithm (WOA) has shown subpar performance in the field of engineering design optimization. However, its simple structure holds significant potential for further development. We aimed to improve WOA, hoping that this variant will match state-of-the-art (SOTA) algorithms in terms of convergence speed and optimization accuracy in numerical optimization tasks. Additionally, we intended for this variant to outperform several SOTA algorithms and the original WOA in the field of engineering design optimization, addressing the shortcomings of WOA in this area and exploring the potential application of WOA in engineering design optimization.

WOA struggles to balance exploration and exploitation, and the population quality tends to deteriorate significantly over iterations, leading to insufficient global exploration and premature convergence to local optima. To address these issues, this paper proposed LSEWOA. LSEWOA introduces the Good Nodes Set method to generate uniformly distributed populations, employs a newly designed Leader-Followers Search-for-Prey Strategy to enhance global exploration, incorporates a novel Spiral-based Encircling Prey Strategy that integrates Spiral flight, utilizes an Enhanced Spiral Updating Strategy combining inertia weight and Tangent flight, and introduces a new update mechanism for the parameter a to better balance exploration and exploitation. Experiments show that LSEWOA effectively addresses the drawbacks of WOA. Furthermore, compared to the classical WOA and other state-of-the-art metaheuristic algorithms, LSEWOA demonstrates significant advantages in both numerical optimization and real-world optimization problems.

## 5. WOA

The Whale Optimization Algorithm (WOA), proposed by Mirjalili et al. in 2016, is a metaheuristic optimization algorithm inspired by the hunting behavior of humpback whales [18]. In WOA, the spiral upward strategy and encircling prey strategy of humpback whales are simulated.

### 5.1. Encircling Prey

Encircling prey behavior is described by Equations (Equation 1) and (Equation 2).(1)$$\vec{D}=\left|\vec{C}\cdot {\vec{X}}^{*}\left(t\right)-\vec{X}\left(t\right)\right|$$(2)$$\vec{X}\left(t+1\right)={\vec{X}}^{*}\left(t\right)-\vec{A}\cdot \vec{D}$$
where *t* is the current iteration; $\vec{A}$ and $\vec{C}$ are coefficient vectors; ${\vec{X}}^{*}\left(t\right)$ is the position of the current best solution; ${\vec{X}}^{*}\left(t\right)$ is the position of the whale.

The results of each iteration are updated if a better solution is found, and the corresponding fitness value improves.

The coefficients $\vec{A}$ and $\vec{C}$ are calculated as follows:(3)$$\vec{A}=2\vec{a}\cdot \vec{r}-\vec{a}$$(4)$$\vec{a}=2-2\cdot \frac{t}{T}$$(5)$$C=2\cdot \vec{r}$$
where $\vec{r}$ is a random vector between 0 and 1; convergence factor *a* decreases linearly from 2 to 0 over the course of iterations, as shown in Figure 1.

### 5.2. Bubble-Net Attacking Method

In addition to encircling prey, whales use a bubble-net attacking method to trap prey by spiraling upward while creating bubbles. This strategy involves two main mechanisms: shrinking encircling and spiral updating.

#### 5.2.1. Shrinking Encircling

This behavior is modeled by decreasing the value of $\vec{A}$ in Equation (Equation 4).

#### 5.2.2. Spiral Updating

The distance between the whale and prey is calculated by Equation (Equation 7), and a spiral equation simulates the upward spiral motion to encircle prey as Equation (Equation 6).(6)$$\vec{X}\left(t+1\right)={\vec{X}}^{*}\left(t\right)+\vec{D^{\prime }}\cdot e^{bl}\cdot \cos\left(2\pi l\right)$$(7)$$\vec{D^{\prime }}=\left|{\vec{X}}^{*}\left(t\right)-\vec{X}\left(t\right)\right|$$(8)$$l=(a_{1}-1)\cdot Rand+1$$(9)$$a_{1}=-1-\frac{t}{T}$$
where *b* is a constant that defines the shape of a logarithmic spiral, usually set to 1; $a_{1}$ is the parameter of the linear change of [−2,−1]; Rand is a random number between 0 and 1; the value of spiral coefficient *l* is [−2,1].

When the WOA algorithm sets the whale’s update position, the encircling strategy and the spiral updating strategy each have a 50% probability, that is:(10)$$\vec{X}\left(t+1\right)=\left\{\begin{array}{cc} {\vec{X}}^{*}\left(t\right)-\vec{A}\cdot \vec{D} & p<0.5\\ {\vec{X}}^{*}\left(t\right)+\vec{D^{\prime }}\cdot e^{bl}\cdot cos\left(2\pi l\right) & p\geq 0.5 \end{array}\right.$$

### 5.3. Searching for Prey

If the whale moves beyond the position where the prey exists, then the whale will abandon the previous moving direction and randomly search for other prey in other directions to avoid falling into a local optimum. This helps avoid local optima and the modelling of the whale searching for prey is as follows:(11)$${\vec{D}}^{\prime \prime }=\left|\vec{C}\cdot {\vec{X}}_{rand}-\vec{X}\left(t\right)\right|$$(12)$$\vec{X}\left(t+1\right)={\vec{X}}_{rand}-\vec{A}\cdot {\vec{D}}^{\prime \prime }$$
where ${\vec{X}}_{rand}$ is a random whale chosen from the current population; $\vec{A}$ and $\vec{C}$ are described in Equations (Equation 3) and (Equation 5).

### 5.4. Population Initialization

Like most metaheuristic algorithms, WOA uses pseudo-random number method for population initialization. The population initialized by pseudo-random number for a population size of *N* = 300, is as shown in Figure 2.(13)$${\vec{X}}_{i,j}=\left(ub-lb\right)\cdot Rand+lb$$
where ${\vec{X}}_{i,j}$ is randomly produced population; ub and lb are the upper limit and lower limit of the problem; Rand is a random number between 0 and 1.

This approach, while simple and direct, often results in poor diversity and uneven distribution of solutions. The phenomenon of population aggregation can easily occur, which can lead to inefficiency in the search process.

### 5.5. Pseudo-Code of WOA

The pseudo-code of WOA is provided in Algorithm 1.
**Algorithm 1** WOA**Begin**Initialize the parameters (T,N,p,etc.);Calculate the fitness of each search agent;The best search agent is ${\vec{X}}^{*}$;    **while** t<T      **for** each search agent         Update *a*, $\vec{A}$, $\vec{C}$, *l*, and *p*;         **if** p<0.5           **if** $|\vec{A}|<1$                           (*Encircling Prey*)              Update the position of the current search agent by Equation (Equation 2);           **else**                           (*Search For Prey*)              Update the position of the current search agent by Equation (Equation 12);           **end if**         **else**                           (*Spiral Updating*)             Update the position of the current search agent by Equation (Equation 7);         **end if**      **end for**      Check if any search agent goes beyond the search space and amend it;      Calculate the fitness of each search agent;      Update ${\vec{X}}^{*}$ if there is a better solution;      t=t+1    **end while****return**${\vec{X}}^{*}$**End**

### 5.6. Advantages and Disadvantages of WOA

It is shown by Algorithm 1 that the structure of WOA is relatively simple, with fewer parameters, making it easy to understand and implement. By dynamically adjusting the convergence factor *a*, WOA balances global exploration in the early stages and local exploitation in the later stages of the iteration, preventing premature convergence to local optima. However, WOA has limitations, particularly in solving complex multi-modal problems, where it may not explore the search space adequately and may converge prematurely. Meanwhile, WOA is suffering an imbalance between exploration and exploitation. Therefore, there is considerable room for improvement in WOA. Hence, we proposed LSEWOA.

## 6. LSEWOA

### 6.1. Good Nodes Set Initialization

The classical WOA uses pseudo-random number method to generate the population, as shown in Figure 2. While simple and effective, this approach often results in poor population diversity and uneven distribution. This leads to inefficient searches, especially when individuals cluster together.

To address these shortcomings, we adopt the Good Nodes Set method for population initialization [28], which ensures a more uniform distribution of solutions. The concept of Good Nodes Set, first introduced by Chinese mathematician Loo-keng Hua, is a method for generating evenly distributed points. This advantage is evident not only in two-dimensional space but also in high-dimensional spaces, as the construction of Good Nodes Set is dimension-independent. Thus, Good Nodes Set Initialization can enhance the quality of whale populations and improve the exploration capabilities of WOA.The population generated by the Good Nodes Set Initialization for a population size of *N* = 300 is shown in Figure 3. In comparison to the pseudo-random number method, the population generated by the Good Nodes Set Initialization is more uniformly distributed, effectively avoiding the phenomenon of individual clustering. Assuming $U^{D}$ is a unit hypercube in a *D*-dimensional Euclidean space, the form of the Good Nodes Set can be described by Equation (Equation 14):(14)$$P_{r}^{M}=\left\{p\left(k\right)=\left(\left\{kr\right\},\left\{kr^{2}\right\},\ldots ,\left\{kr^{D}\right\}\right)|k=1,2,\ldots ,M\right\}$$
where {*x*} represents the fractional part of *x*; *M* is the number of nodes; *r* is a deviation parameter greater than zero; the constant C(r,ε) is associated only with *r* and ε is related to and is a constant greater than zero.

This set $P_{r}^{M}$ is called Good Nodes Set and each node p(k) in it is called a Good Node. Assume that the upper and lower bounds of the $i^{th}$ dimension of the search space ${\vec{X}}_{max}^{i}$ are and ${\vec{X}}_{min}^{i}$, then the mapping formula for mapping the Good Nodes Set to the actual search space is:(15)$${\vec{X}}_{k}^{i}={\vec{X}}_{min}^{i}+p_{i}\left(k\right)\cdot \left({\vec{X}}_{max}^{i}-{\vec{X}}_{min}^{i}\right)$$

### 6.2. Leader-Followers Search-for-Prey Strategy

The search-for-prey strategy of the original WOA by randomly selecting a whale individual, increasing the diversity of WOA to some extent. However, it also leads to unstable convergence, making the search paths of individuals lack clear direction and regularity. Particularly in later iterations, the search strategy may become overly reliant on randomness, resulting in premature convergence or trapping in local optima. Additionally, the search-for-prey strategy in WOA overly depends on randomly selected individuals to update positions, which may lead to insufficient utilization of the population’s information. To address these issues, this paper mimics the behavior of the leader guiding followers towards the prey during the whale’s foraging process and proposes the Leader-Followers Search-for-Prey Strategy. The detailed process is shown in Figure 4. This strategy aims to resolve the drawbacks of the WOA exploration phase, such as insufficient utilization of population information and excessive reliance on randomness in the search process. The modeling of the Leader-Followers Search-for-Prey Strategy is as follows:(16)$$\vec{X}\left(t+1\right)=\varepsilon \cdot {\vec{X}}^{*}\left(t\right)+\left|{\vec{X}}_{R}\left(t\right)-{\vec{X}}^{*}\left(t\right)\right|$$(17)$$\varepsilon =1-\frac{t}{T}$$(18)$${\vec{X}}_{R}\left(t\right)=\frac{1}{N}\sum_{i=1}^{N}{\vec{X}}_{i}\left(t\right)$$
where ε is the attraction coefficient of the Leader, and its calculation formula is shown in Equation (Equation 17); *t* is the current iteration number; *T* is the maximum number of iterations; ${\vec{X}}_{i}$ represents the position of the whale individual; ${\vec{X}}_{R}$ is the average position of all whale individuals, calculated as shown in Equation (Equation 18); *N* represents the population size; ${\vec{X}}^{*}$ denotes the position of the current best solution.

The Leader-Followers Search-for-Prey Strategy updates the position of whale individuals by using the position of the current best whale and the average position of all whales. This approach makes better use of the collective information of the population, allowing the movement of whale individuals to be more based on the population structure characteristics rather than purely random behavior, thus avoiding too little or too much mutual influence between whale individuals. By leveraging the dominant influence of the Leader and combining the population average, the position update of each individual is attracted by the global optimal position while being constrained by the population distribution, thus expanding the search range of the algorithm. This helps to make the search more targeted, reduces randomness, and, by decreasing ε, gradually reduces the update intensity, enabling the algorithm to quickly identify potential solutions in a larger search space. In the early stages, when ε is larger, the Leader’s attraction to the Followers is stronger, enhancing the exploratory behavior. In later stages, this attraction weakens, promoting more exploitative behavior, thus helping the algorithm to better balance exploration and exploitation. The Leader’s strong guidance helps the whale population to form a clear global search direction, laying the foundation for later development. In the early stages, the positions of the whale individuals are more dispersed. If the dependency on the Leader is small, the updates between individuals may lack concentration, leading to disordered search behavior. In later stages, when ε is smaller, the weaker attraction from the Leader prevents the whale population from prematurely converging around the Leader, which helps preserve population diversity. During each position update, whale individuals, by focusing on the relationship between themselves and the population average, can perform fine searches within their neighborhood with smaller steps, contributing to greater convergence stability and avoiding large fluctuations in position, thereby improving the accuracy of the final solution. Additionally, the linear variation of ε makes the search strategy smoother and more gradual, helping to reduce unnecessary fluctuations during convergence and improving both the convergence accuracy and speed of the algorithm.

### 6.3. Spiral-Based Encircling Prey Strategy

In the encircling prey phase of the original WOA, the position update method is based on the Euclidean distance between the current best solution and the whale’s individual position. This linear shrinking strategy often leads to a monotonous search behavior, lacking flexibility, and may get trapped in local optima, especially in later stages. Additionally, when the value of *A* is small, the search range is restricted, which could prevent the algorithm from escaping local optima. Inspired by the Spiral flight mechanism, this paper proposes a Spiral-based Encircling Prey Strategy. The modeling of the Spiral-based Encircling Prey Strategy is shown in Equations (Equation 19)–(Equation 22). Figure 5 illustrates the concept of the Spiral-based Encircling Prey Strategy.(19)$$\vec{X}\left(t+1\right)={\vec{X}}^{*}\left(t\right)+e^{Z\cdot L}\cdot cos\left(2\pi L\right)\cdot \left|\vec{A}\cdot \vec{D}\right|$$(20)$$\vec{D}=\left|\vec{C}\cdot {\vec{X}}^{*}\left(t\right)-\vec{X}\left(t\right)\right|$$(21)L=2·r−1(22)$$Z=e^{k\cdot cos(\pi \cdot \left(1-\frac{t}{T}\right))}$$
where $\vec{A}$ and $\vec{C}$ are coefficient vectors; *Z* represents the Spiral flight step size; *s* and *k* are spiral coefficients; and *r* is a random number between 0 and 1.

The Spiral-based Encircling Prey Strategy increases the randomness and nonlinear variation of position updates by introducing the nonlinear step size *Z* of Spiral flight. This enables whale individuals to take diversified paths when approaching the prey. During the process of approaching the prey, the whale individuals can explore the local space more thoroughly, and when trapped in a local optimum, they have a higher probability of escaping and finding a better solution. This strategy helps avoid premature convergence in complex problems.

### 6.4. Enhanced Spiral Updating Strategy

The Spiral Updating Strategy in the original WOA helps with local search, but due to the lack of randomness and disturbance mechanisms, the position update of the whale individuals consistently depends on the Leader’s position at the same level. This results in a lack of diversity and the ability to escape from local optima during the development phase, making it prone to getting trapped in local optimum solutions. Therefore, in this paper, inertia weight and Tangent flight were introduced into the original Spiral Updating Strategy, and an Enhanced Spiral Updating Strategy is proposed.

#### 6.4.1. Inertia Weight ω

The concept of inertia weight ω was first introduced by Shi et al. in the PSO [29], and it led to significant performance improvements. Later, numerous studies demonstrated that inertia weight is effective in balancing global exploration and local exploitation. Therefore, an inertia weight ω based on the Sigmoid function is introduced into the prey capture strategy of the WOA, as shown in Equation (Equation 23). Figure 6 compares the common inertia weights with the inertia weight proposed in this paper with $k_{1}$ = 10, $k_{1}$ = 15 and $k_{1}$ = 20. This inertia weight ω increases from 0 to 0.9, initially increasing slowly, then rapidly increasing in the middle phase, and finally slowing down towards the later stage. Furthermore, as $k_{1}$ increases, the speed of change in the middle phase becomes faster.(23)$$\omega =\frac{0.9}{1+e^{-k_{1}\left(\frac{t}{T}-0.5\right)}}$$
where *t* is the current iteration; *T* is the maximum number of iterations; $k_{1}$ is the slope parameter of the Sigmoid function and the value of the parameter $k_{1}$ will be discussed.

In the early stages of iteration, the inertia weight ω is small, and when updating the positions, the distance $D^{\prime }$ between the whale individual and the leader has a lower reference value. As a result, the whale individual is less influenced by the leader’s attraction, allowing for greater freedom to perform global exploration. In the later stages of iteration, when the inertia weight is larger, the whale individual is more strongly influenced by the leader’s attraction. At this point, the freedom of the whale individual is restricted, and the whale individual closely follows the leader for detailed local exploitation.

#### 6.4.2. Tangent Flight

Tangent flight is a new step-size calculation method based on the tangent function, proposed in 2021 by the Tangent Search Algorithm (TSA) [30]. Tangent flight is based on a heavy-tailed distribution and is characterized by large step-size movements. Figure 7 shows a simulation of tangent flight. The following is the calculation method for the Tangent flight step size.(24)$$T_{f}=\tan\left(\theta \right)$$(25)$$\theta =rand\cdot \frac{\pi }{2}$$
where $T_{f}$ is the step size of Tangent flight; rand is a random number from 0 to 1.

Larger Tangent flight steps are beneficial for exploration, while smaller Tangent flight steps are advantageous for exploitation. From the formula, the value of θ ranges from 0 to $\frac{\pi }{2}$. The closer θ is to $\frac{\pi }{2}$, the larger $T_{f}$ becomes, which corresponds to a larger step size in Tangent flight, favoring global exploration. On the other hand, the closer θ is to 0, the smaller $tanT_{f}$ becomes, resulting in smaller step size movements, which are suitable for local exploitation. Compared to Levy flight, the frequency of large steps in Tangent flight is higher, which compensates for the issue of search distances being too large or too small in Levy flight. Tangent flight is more advantageous than Levy flight in escaping local optima. Tangent flight enhances the global exploration ability of the WOA while effectively addressing its local exploitation shortcomings. This, in turn, accelerates WOA’s convergence rate and improves overall performance.

### 6.5. Calculation of Enhanced Spiral Updating Strategy

Enhanced Spiral Updating Strategy combining inertial weight and step size of tangent flight is modeled as follows:(26)$$\vec{X}\left(t+1\right)={\vec{X}}^{*}\left(t\right)\cdot T_{f}+\omega \cdot {\vec{D}}^{\prime }\cdot e^{bl}\cdot \cos\left(2\pi l\right)$$
where $T_{f}$ is the step size of Tangent flight; ω is the proposed inertia weight.

The Enhanced Spiral Updating Strategy enhances the diversity of the whale individual’s search. The original spiral ascent strategy itself, through the combination of exponential functions and cosine waves, makes the whale individual’s search trajectory more complex and directional. After introducing the inertial weight ω, the amplitude and jump range of the trajectory are dynamically adjusted with each iteration, further enriching the whale’s search path. By combining the step size $T_{f}$ of Tangent flight with heavy-tail characteristics, the whale individual’s movement ability is expanded globally, allowing for large random jumps. While accelerating the algorithm’s convergence speed, the larger step sizes provide WOA with an effective ability to escape local optima, making it more efficient in solving complex optimization problems.

### 6.6. Redesigned Convergence Factor *a*

This paper proposed a new updating method of the convergence factor *a* based on Sigmoid function to balance global exploration and local exploitation, with its calculation method shown in Equation (Equation 27). Figure 8 compares the proposed approach with $k_{2}$ = 15, $k_{2}$ = 20 and $k_{2}$ = 25 to the original linear decay method in the classical WOA. The update strategy results in a slower decrease in the early stage, a rapid reduction in the middle stage, and a slower decrease again in the later stage. Moreover, as $k_{2}$ increases, the speed of change in the middle stage becomes faster. This approach simulates a more complex variation process, giving the algorithm different convergence characteristics at different stages: the slower reduction in the early stage helps to broadly explore the search space; the rapid decrease in the middle stage accelerates convergence; and the slower reduction in the later stage preserves a certain level of exploration ability while enhancing local exploitation, facilitating more refined local search. This strategy provides a better balance between global and local search, offers more flexible convergence behavior, alleviates premature convergence, increases adaptability to different search environments, and improves the precision of the final solution. These improvements make WOA more efficient and reliable when dealing with complex optimization problems.(27)$$a=2-\frac{2}{1+e^{-k_{2}\left(\frac{t}{T}-0.5\right)}}$$
where *t* is the current iteration count; *T* is the maximum number of iterations; $k_{2}$ is the slope parameter of the Sigmoid function and the value of the parameter $k_{2}$ will be discussed.

### 6.7. Pseudo-Code of LSEWOA

The pseudo-code of LSEWOA is provided in Algorithm 2.
**Algorithm 2** LSEWOA**Begin**Initialize the parameters (T,N,p,etc.);Initialize population using Good Nodes Set Initialization;Calculate the fitness of each search agent;The best search agent is ${\vec{X}}^{*}$;    **while** t<T      **for** each search agent         Update *a*, ω, ε, $\vec{A}$, $\vec{C}$, *l*, and *p*;         **if** p<0.5           **if** $|\vec{A}|<1$                              (*Spiral-based Encircling Prey*)              Update the position of the current search agent by Equation (Equation 19);           **else**                           (*Leader-Followers Search-for-Prey*)              Update the position of the current search agent by Equation (Equation 16);           **end if**         **else**                          (*Enhanced Spiral Updating Strategy*)             Update the position of the current search agent by Equation (Equation 26);         **end if**      **end for**      Check if any search agent goes beyond the search space and amend it;      Calculate the fitness of each search agent;      Update ${\vec{X}}^{*}$ if there is a better solution;      t=t+1    **end while****return**${\vec{X}}^{*}$**End**

### 6.8. Time Complexity Analysis

#### 6.8.1. Time Complexity of WOA

Assume that the time complexity of initialization in WOA is O(ND). During each iteration, the time complexity of boundary checking is O(ND), the time complexity of fitness evaluation is O(ND), and the total time complexity of position updates is O(ND). Therefore, the total time complexity per iteration is O(ND). If the algorithm iterates *T* times, the total time complexity is calculated as:

Total Time Complexity1 = Initialization + *T* ∗ (the total time complexity per iteration) = O(ND) + *T* ∗ O(ND) = O(T∗ND)

#### 6.8.2. Time Complexity of LSEWOA

Assume that the time complexity of initialization in LSEWOA is O(ND). During each iteration, the time complexity of boundary checking is O(ND), the time complexity of fitness evaluation is O(ND), and the total time complexity of position updates is O(ND). Therefore, the total time complexity per iteration is O(ND). If the algorithm iterates *T* times, the total time complexity is calculated as:

Total Time Complexity2 = Initialization + *T* ∗ (the total time complexity per iteration) = O(ND) + *T* ∗ O(ND) = O(T∗ND)

In summary, the time complexity of LSEWOA and WOA are the same, both are O(T∗ND).

## 7. Experiments

The experimental setup for this study includes a Windows 11 (64-bit) operating system, an Intel(R) Core(TM) i5-8300H CPU @ 2.30 GHz processor, and 8 GB of RAM. The simulation platform used is MATLAB R2023a. The algorithm was tested on 23 classical benchmark functions in CEC2005 test suit in Table 3 and Table A1 [31]. To verify the performance and effectiveness of LSEWOA, the following experiments were conducted.

A parameter sensitivity analysis experiment was performed on different LSEWOAs with various $k_{1}$ and $k_{2}$, aiming at choosing the perfect option of $k_{1}$ and $k_{2}$ for parameter *a* and inertia weight ω respectively to better balance exploration and exploitation.A qualitative analysis experiment was performed by applying LSEWOA on the 23 benchmark functions to comprehensively evaluate the performance, robustness and exploration-exploitation balance of LSEWOA in different types of problems, by assessing search behavior, exploration-exploitation capability and population diversity.An ablation study was performed by removing each of the five improvement strategies from LSEWOA and testing on 23 benchmark functions.LSEWOA was tested against five excellent WOA variants on the benchmark functions.LSEWOA was compared with the canonical WOA and several state-of-the-art algorithms on the benchmark functions.

### 7.1. Parameter Sensitivity Analysis Experiment

As shown in Equations (Equation 23) and (Equation 27), the values of ω with different $k_{1}$ and parameter *a* with different $k_{2}$ can significantly affect the performance of LSEWOA. The ω with different $k_{1}$ values has a substantial impact on the change in the degree of dependency of whale individuals on the leader during the position update in the Enhanced Spiral Updating Strategy. Meanwhile, as parameter *a* influences the balance between exploration and exploitation, it is necessary to explore the shape of the Sigmoid function of parameter *a*. In this experiment, we discuss the values of $k_{1}$ and $k_{2}$. Specifically, we performed the Friedman test on LSEWOA with different combinations of $k_{1}$ and $k_{2}$ over 23 benchmark functions. The number of iterations was set to *T* = 500, and the population size was set to *N* = 30. Each version of LSEWOA with different $k_{1}$ and $k_{2}$ was executed 30 times on the 23 benchmark functions listed in Table 3. The Friedman values are recorded in Table 4. ‘**LSEWOA(20, 15)**’ means $k_{1}$ = 20, $k_{2}$ = 15; ‘**LSEWOA(15, 20)**’ means $k_{1}$ = 15, $k_{2}$ = 20. ‘**Average**’ means average Friedman value.

The results indicate that LSEWOA with $k_{1}$ = 20 and $k_{2}$ = 25 performed the best. Notably, the Friedman value for LSEWOA with $k_{1}$ = 15 and $k_{2}$ = 25 was 4.0913, for $k_{1}$ = 20 and $k_{2}$ = 25 it was 3.7935, and for $k_{1}$ = 25 and $k_{2}$ = 25 it was 3.9443. This confirms the importance of exploring the values of $k_{1}$ and $k_{2}$.

### 7.2. Ablation Study

In this ablation study, we removed five improvement strategies from LSEWOA individually:LSEWOA1: We replaced the Good Nodes Set Initialization with pseudo-random number initialization in LSEWOA, which is referred to as LSEWOA1;LSEWOA2: We replaced the Leader-Followers Search-for-Prey Strategy with the original WOA’s Search-for-prey strategy, referred to as LSEWOA2;LSEWOA3: We replaced the Spiral-based Encircling Prey Strategy with the original WOA’s encircling prey strategy, referred to as LSEWOA3;LSEWOA4: We replaced the Enhanced Spiral Updating Strategy with the original WOA’s spiral updating mechanism, referred to as LSEWOA4;LSEWOA5: We replaced the proposed update mechanism of parameter *a* with the one in classical WOA, referred to as LSEWOA5.

Additionally, we set the number of iterations to *T* = 500 and the population size to *N* = 30. Each algorithm was run 30 times independently on 23 benchmark functions in 30 dimensions to validate the effectiveness of each improvement strategy. The iteration curves are shown in Figure 9.

As seen in the figure, the Good Nodes Set Initialization generates a more uniformly distributed whale population, allowing whale individuals to explore the solution space more effectively. This initialization strategy demonstrates a significant advantage in handling complex multi-modal problems such as F12–F15. The Leader-Followers Search-for-Prey Strategy uses the average position of whale individuals and the leader’s distance as one of the references for position updates, fully utilizing the information within the population. Furthermore, with the leader’s decaying attraction, this strategy better guides the followers toward the prey, enabling the algorithm to balance exploration and exploitation more naturally throughout the iterations, thus improving the convergence speed for functions like F1–F4. The Spiral-based Encircling Prey Strategy, by introducing a nonlinear step size through spiral flight, increases the randomness and nonlinear variation of position updates, introducing a certain degree of periodicity and randomness into the algorithm. This allows the algorithm to continuously escape local optima, thus preventing premature convergence when tackling complex problems like F5–F6 and F12–F13. The Enhanced Spiral Updating Strategy further enriches the whale’s search path. While accelerating the algorithm’s convergence speed, the larger step size enables WOA to effectively escape local optima, greatly improving the convergence speed on functions like F1–F4, F9–F11. Additionally, the Sigmoid function-based update for parameter a, proposed in this paper, provides WOA with a better ability to balance global exploration and local exploitation, further improving the solution accuracy. This allows WOA to focus more effectively on local exploitation in the later iterations, continuously searching for better solutions.

### 7.3. Qualitative Analysis Experiment

In the qualitative analysis experiment, we set the number of iterations to *T* = 500 and the population size to *N* = 30, and ran LSEWOA independently on 23 benchmark functions in 30 dimensions in Table 3 to analyze the search history, exploration-exploitation ratio, and population diversity of LSEWOA. In addition, we have provided the landscape of the benchmark functions and the iteration curves for reference. The results of the qualitative analysis are shown in Figure 10, Figure 11, Figure 12, which includes:the landscape of benchmark functions;the search history of the whale population;the exploration-exploitation ratio;the changes in population diversity;the iteration curves.

The search history represents the positions and distribution of the whale individuals. In the figures of the search history of the whale population, red circles indicate the location of the global optimum, and blue circles represent the search history of the whale individuals. It is noteworthy that LSEWOA effectively explores the entire search space, as indicated by the positions of the whale individuals. LSEWOA demonstrates a rapid convergence speed in single-modal functions such as F1–F4, where whale individuals find the optimal solution within a limited number of iterations, resulting in a concentrated distribution of individuals within the solution space. In the case of complex multi-modal functions like F8, which have many local optima, LSEWOA first conducts a quick global exploration and then performs detailed exploitation in promising regions. During the exploitation phase, the oscillation term in the Spiral-based Encircling Prey Strategy and the tangent flight step size in the Enhanced Spiral Updating Strategy allow LSEWOA to not only converge rapidly but also effectively escape from local optima, thereby exploring more promising solutions.

In terms of balancing exploration and exploitation, LSEWOA performs excellently. When dealing with uni-modal functions such as F1–F4, the exploitation ratio of LSEWOA increases rapidly in the early iterations, indicating strong exploitation ability. When handling multi-modal functions like F14–F15 and composite functions like F21–F23, LSEWOA exhibits a higher exploration ratio in the early iterations, demonstrating its strong global exploration capability and its ability to continuously identify more promising regions in the search space.

In functions F17–F23, the population diversity curve of LSEWOA consistently fluctuates and maintains a high value. This indicates that LSEWOA is able to maintain high population diversity when handling complex multi-modal functions, effectively preventing premature convergence caused by the population clustering in certain areas.

### 7.4. Comparative Experiment with State-of-the-Art WOA Variants

To evaluate the superiority of LSEWOA, we selected following five state-of-the-art WOA variants as controls and tested them with LSEWOA on the 23 benchmark functions in 30 dimensions listed in Table 3.

WOAV1: The WOA variant (eWOA) that introduces adaptive parameter adjustment, multi-strategy search mechanisms, and elite retention strategies is referred to as WOAV1 [32];WOAV2: The WOA variant (NHWOA) that incorporates multiple subpopulations, dynamically adjusted control parameters, adaptive position update mechanisms, and Levy flight perturbations is referred to as WOAV2 [33];WOAV3: The WOA variant (MSWOA) that introduces adaptive weights, dynamic convergence factors, and Levy flight is referred to as WOAV3 [34];WOAV4: The WOA variant (MWOA) that uses an iteration-based cosine function and exponential decay adjustment for parameters, hybrid mutation strategies, Levy flight, and hybrid update mechanisms is referred to as WOAV4 [35];WOAV5: The WOA variant (WOA_LFDE) that introduces Levy flight and Differential Evolution strategies is referred to as WOAV5 [36].

We uniformly set the number of iterations to *T* = 500 and the population size to *N* = 30. Each of these five WOA variants and LSEWOA was run 30 times on the benchmark functions, recording the average fitness (Ave), standard deviation (Std), *p*-values of Wilcoxon rank-sum test, and Friedman values for performance analysis. And finally we evaluate the overall effectiveness (OE). The iteration curves are shown in Figure 13 and Table 5, Table 6, Table 7.

#### 7.4.1. Parametric Analysis

The experimental results show that LSEWOA performed excellently in this comparative experiment. LSEWOA outperformed all other algorithms in terms of both mean and standard deviation on F1–F17 and F20–F23. In F18–F19, although LSEWOA has the optimal mean value, its stability is lower than that of other variants.

In algorithm performance evaluation, the average fitness and standard deviation are commonly used to measure convergence and stability, but they do not provide an intuitive representation of algorithm performance. Relying solely on the average fitness and standard deviation to compare different algorithms has limitations, which is why non-parametric tests, such as the Wilcoxon rank-sum test and Friedman test, are often introduced. These statistical tests provide deeper analysis and reliability verification.

#### 7.4.2. Non-Parametric Wilcoxon Rank-Sum Test

The Wilcoxon rank-sum test is a non-parametric method used to compare whether there are significant differences between the distributions of two independent samples. The Wilcoxon rank-sum test avoids the bias that may arise from relying solely on mean and variance, is less sensitive to outliers, and can provide more robust performance evaluation than mean and standard deviation. In the comparison of optimization algorithms, when we want to compare whether the effects of two algorithms show significant differences, the Wilcoxon test helps us determine whether this difference is statistically significant. If the *p*-value from the Wilcoxon test is smaller than the set significance level (usually 0.05), we can conclude that the performance difference between the two algorithms is significant, rather than being caused by random error. In the Wilcoxon rank-sum test, LSEWOA shown significant differences with WOAV2 and WOAV5 across all benchmark functions. LSEWOA did not show significant differences with WOAV1 and WOAV4 in F1 and F9–F11 because, in these functions, both algorithms quickly converge to the optimal solution. Similarly, LSEWOA did not show significant differences with WOAV3 in F10–F11, as WOAV3 also converged rapidly on these functions. LSEWOA did not show significant differences with WOAV4 in F3, as both algorithms quickly reached the optimal solution in this function.

#### 7.4.3. Non-Parametric Friedman Test

The Friedman test is also a non-parametric method used to compare differences among three or more related samples. It is a non-parametric version of repeated-measures ANOVA and is suitable for scenarios where different algorithms are repeatedly tested on the same dataset. The Friedman test can identify statistically significant differences among algorithms. By comparing the performance of multiple algorithms across multiple datasets or test environments, the Friedman test effectively eliminates sample bias and provides a fairer comparison. In the Friedman test, LSEWOA had an average Friedman value of 1.6710, ranking first, followed by WOAV1, WOAV2, and WOAV3 in second, third, and fourth places, respectively. WOAV5 and WOAV4 ranked fifth and sixth.

### 7.5. Scalability Experiment of LSWOA

In the 23 benchmark functions, F1–F13 are functions with expandable dimensions, while F14–F23 are functions with fixed dimensions. To validate the ability of LSEWOA to handle problems of different dimensions and complexities, this experiment expands the dimensions of the expandable functions (F1–F13) to 50 and 100 dimensions, while keeping the dimensions of the functions (F14–F23) fixed. LSEWOA was compared with WOAV1, WOAV2, WOAV3, WOAV4, and WOAV5 on the 23 benchmark functions on higher dimension of 50 (*D* = 50) and 100 (*D* = 100). The comparison results are shown in Table 8. The results shown that LSEWOA performed excellently in the scalability comparison experiment. LSEWOA had a significant advantage in handling problems with different dimensions.

#### Overall Effectiveness of LSEWOA

Table 7 summarizes all performance results of LSEWOA and other algorithms by a useful metric named overall effectiveness (OE). In Table 7, *w* indicates win, *t* indicates tie and *l* indicates loss. The OE of each algorithm is computed by Equation (Equation 28) [37].(28)$$OE=\frac{N-L}{L}\cdot 100$$
where *N* is the total number of tests; *L* is the total number of losing tests for each algorithm.

The results reveal that LSEWOA with overall effectiveness of 97.10% was the most effective algorithm. In summary, LSEWOA demonstrated exceptional performance on the classical benchmark functions and shown clear differences compared to the five selected WOA variants.

### 7.6. Comparative Experiment with State-of-the-Art Metaheuristic Algorithms in 30 Dimension

To further validate the effectiveness of LSEWOA, this study compares it with several state-of-the-art metaheuristic algorithms in 30 dimensions, including the Grey Wolf Optimizer (GWO) [38], Harris Hawk Optimization algorithm (HHO) [7], Zebra Optimization Algorithm (ZOA) [39], Slime Mould Algorithm (SMA) [40], Sine Cosine Algorithm (SCA) [41], Attraction-Repulsion Optimization Algorithm (AROA) [42], Rime Optimization Algorithm (RIME) [43], and Whale Optimization Algorithm (WOA) [18], on the benchmark functions listed in Table 9. The parameter settings for each algorithm are shown in Table 10. The number of iterations was uniformly set to *T* = 500, with a population size of *N* = 30. Each algorithm was independently run 30 times on 23 benchmark functions, and the average fitness (Ave), standard deviation (Std), *p*-values of Wilcoxon rank-sum test, and Friedman values were recorded for performance analysis. The experimental results are shown in Figure 14 and Table 11, Table 12, Table 13.

The experimental results demonstrate that LSEWOA achieved the best overall performance among all the algorithms in 30 dimensions and shown a significant improvement in overall performance compared to the original WOA. As shown in Figure 14 and Table 11, LSEWOA outperformed all algorithms in terms of both the average fitness and standard deviation when solving F1–F17 and F20–F23. However, in solving F18–F19, the stability of LSEWOA was lower than that of SMA. For most functions, LSEWOA quickly found the optimal solution, exhibiting a faster convergence rate and higher solution accuracy. This confirms that LSEWOA had good adaptability and robustness when handling different types of problems.

In the Wilcoxon rank-sum test, as shown in Table 12, LSEWOA shown significant differences from GWO, SCA, AROA, and RIME across all test functions. LSEWOA did not show significant differences from HHO, ZOA, and SMA on F9-F11 because all of these algorithms quickly found the optimal values for these three functions within the given number of iterations. LSEWOA also did not show significant differences from RIME on F19.

Ranking the average Friedman values of each algorithm, as shown in Table 12, LSEWOA had the lowest average Friedman value of 1.5587, ranking first. SMA had an average Friedman value of 2.7565, ranking second. ZOA and HHO had average Friedman values of 3.8956 and 4.9297, ranking third and fourth, respectively. GWO, WOA, and RIME had average Friedman values of 5.2312, 5.2587, and 6.0000, ranking fifth, sixth, and seventh, respectively. AROA and SCA had average Friedman values of 7.2913 and 8.0783, ranking eighth and ninth.

### 7.7. Comparative Experiment with State-of-the-Art Metaheuristic Algorithms in Higher Dimensions

The parameter settings for each algorithm were shown in Table 10. The number of iterations was set to *T* = 500, and the population size was set to *N* = 30. Each algorithm was run independently 30 times on the benchmark functions in 50 and 100 dimensions, with the *p*-value from Wilcoxon rank-sum test and Friedman values recorded for performance analysis. The experimental results were shown in Table 14.

The results shown that LSEWOA performed excellently in the scalability comparison experiment. Table 13 reveals that LSEWOA with overall effectiveness of 97.10% was the most effective algorithm. In summary, LSEWOA exhibited the best overall performance among all the algorithms, demonstrating strong competitiveness compared to other state-of-the-art metaheuristic algorithms.

## 8. Engineering Optimization

In this section, nine engineering design problems will be used to test the superior performance of the developed LSEWOA in solving various practical applications. LSEWOA will be compared with GWO, HHO, ZOA, SMA, SCA, AROA, RIME, and WOA. The parameter settings for each algorithm are shown in Table 10. The iteration number is uniformly set to *T* = 500 and the population size to *N* = 30. Each algorithm is run 30 times independently on each engineering design optimization problem, with the average fitness (Ave) and standard deviation (Std) recorded for performance analysis.

### 8.1. Three-Bar Truss

The three-bar truss is a simple structural system consisting of three members, as shown in Figure 15. It is commonly used to support concentrated loads and is widely applied in engineering fields such as bridges, buildings, and aerospace. The three-bar truss design problem is a classic structural optimization problem, often used to study the mechanical behavior of simple structures under external loading conditions. In the three-bar truss design problem, the objective is to optimize the cross-sectional areas of the truss members to minimize material usage while ensuring that the structure meets the required mechanical performance.

This optimization problem involves a nonlinear objective function, three nonlinear inequality constraints, and two continuous decision variables $x_{1}$ and $x_{2}$. The objective function for the three-bar truss design problem can be described as follows:


*Variable:*

$$\begin{array}{c} x=[x_{1},x_{2}] \end{array}$$




*Minimize:*

(29)
$$f\left(x\right)=\left(2\sqrt{2}x_{1}+x_{2}\right)\cdot l$$




*Subject to:*

(30)
$$g_{1}\left(x\right)=\frac{\sqrt{2}x_{1}+x_{2}}{\sqrt{2}\cdot x_{1}^{2}+2x_{1}\cdot x_{2}}P-\sigma \leq 0$$


(31)
$$g_{2}\left(x\right)=\frac{x_{2}}{\sqrt{2}x_{1}^{2}+2x_{1}\cdot x_{2}}P-\sigma \leq 0$$


(32)
$$g_{3}\left(x\right)=\frac{1}{\sqrt{2}x_{2}+x_{1}}P-\sigma \leq 0$$




*Where:*

$$\begin{array}{c} l=100\;\mathrm{cm};\;P=2\;\mathrm{kN}/{\mathrm{cm}}^{2};\;\sigma =2\;\mathrm{kN}/{\mathrm{cm}}^{2} \end{array}$$




*Variable range:*

$$\begin{array}{c} 0\leq x_{i}\leq 1,\;i=1,2 \end{array}$$



The experimental results are shown in Figure 16 and Table 15. From Table 15, it can be observed that LSEWOA significantly outperforms other algorithms in terms of stability, with the best optimization accuracy among all algorithms. This indicates that LSEWOA has a significant advantage when handling such optimization problems.

### 8.2. Tension/Compression Spring

The extension/compression spring, as shown in Figure 17, plays a crucial role in modern industry, with widespread applications in fields such as automotive, home appliances, and electronics. Its design optimization not only helps improve product performance and extend service life, but also reduces costs and enhances manufacturing efficiency. Through reasonable design optimization, the spring can achieve optimal performance in dynamic working environments and meet various stringent requirements. The optimization objective of the design problem for the extension/compression spring is the minimization of its mass. The problem needs to be solved under constraints such as shear force, deflection, fluctuation frequency, and outer diameter. There are three design variables in this problem: coil diameter *d*, mean coil diameter *D*, and number of coils *N*. There are also four constraints, $g_{1}$ to $g_{4}$. The mathematical model of the problem is as follows,


*Variable:*

$$\begin{array}{c} x=\left[d,D,N\right]=\left[x_{1},x_{2},x_{3}\right] \end{array}$$




*Minimize:*

(33)
$$f\left(x\right)=\left(x_{3}+2\right)\cdot x_{2}\cdot x_{1}^{2}$$




*Subject to:*

(34)
$$g_{1}\left(x\right)=1-\frac{x_{2}^{3}\cdot x_{3}}{71785x_{1}^{4}}\leq 0$$


(35)
$$g_{2}\left(x\right)=\frac{4x_{2}^{2}-x_{1}\cdot x_{2}}{12566(x_{2}\cdot x_{1}^{3}-x_{4})}+\frac{1}{5108x_{1}^{2}}-1\leq 0$$


(36)
$$g_{3}\left(x\right)=1-\frac{140.45x_{1}}{x_{2}^{2}\cdot x_{3}}\leq 0$$


(37)
$$g_{4}\left(x\right)=\frac{x_{1}+x_{2}}{1.5}-1\leq 0$$




*Variable range:*

$$\begin{array}{c} 0.05\leq x_{1}\leq 2,\;0.25\leq x_{2}\leq 1.3,\;2.0\leq x_{3}\leq 15 \end{array}$$



The experimental results are presented in Figure 18 and Table 15. As shown in Table 15, the stability of LSEWOA in the Tension/Compression Spring design problem significantly surpasses other algorithms, and it achieves the highest optimization accuracy among all algorithms. This indicates that LSEWOA has a significant advantage in handling such optimization problems.

### 8.3. Speed Reducer

A speed reducer is a mechanical transmission device and one of the key components of a gearbox, shown in Figure 19. It is primarily used to reduce the rotational speed of an electric motor or other power sources while increasing the output torque. The reducer achieves this speed reduction through gears, worm gears, or other transmission mechanisms. It is typically applied in situations where there is a need to decrease the rotational speed, increase torque, or adjust the direction of motion.

In the optimization design of a reducer, the goal is to minimize the weight of the reducer. This problem involves seven variables, which are as follows: the width of the gear teeth $x_{1}$, the gear module $x_{2}$, the number of teeth on the small gear $x_{3}$, the length of the first shaft between the bearings $x_{4}$, the length of the second shaft between the bearings $x_{5}$, the diameter of the first shaft $x_{6}$, and the diameter of the second shaft $x_{7}$. Furthermore, this problem also involves eleven constraints, $g_{1}$ to $g_{11}$. The mathematical formulation of the problem is as follows,


*Variable:*

$$\begin{array}{c} x=[x_{1},x_{2},x_{3},x_{4},x_{5},x_{6},x_{7}] \end{array}$$




*Minimize:*

(38)
y=f(x)




*Subject to:*

(39)
$$g_{1}=\frac{27}{x_{1}\cdot x_{2}^{2}\cdot x_{3}}-1\leq 0;$$


(40)
$$g_{2}=\frac{397.5}{x_{1}\cdot x_{2}^{2}\cdot x_{3}^{2}}-1\leq 0;$$


(41)
$$g_{3}=\frac{1.93x_{4}^{3}}{x_{2}\cdot x_{6}^{4}\cdot x_{3}}-1\leq 0;$$


(42)
$$g_{4}=\frac{1.93x_{5}^{3}}{x_{2}\cdot x_{7}^{4}\cdot x_{3}}-1\leq 0;$$


(43)
$$g_{5}=\frac{\sqrt{16.91\cdot 10^{6}+{\left(\frac{745x_{4}}{x_{2}\cdot x_{3}}\right)}^{2}}}{110x_{6}^{3}}-1\leq 0;$$


(44)
$$g_{6}=\frac{\sqrt{157.5\cdot 10^{6}+{\left(\frac{745x_{4}}{x_{2}\cdot x_{3}}\right)}^{2}}}{85x_{7}^{3}}-1\leq 0;$$


(45)
$$g_{7}=\frac{x_{2}\cdot x_{3}}{40}-1\leq 0;$$


(46)
$$g_{8}=\frac{5x_{2}}{x_{1}}-1\leq 0;$$


(47)
$$g_{9}=\frac{x_{1}}{12x_{2}}-1\leq 0;$$


(48)
$$g_{10}=\frac{1.5x_{6}+1.9}{x_{4}}-1\leq 0;$$


(49)
$$g_{11}=\frac{1.1x_{7}+1.9}{x_{5}}-1\leq 0;$$




*Variable range:*

$$2.6\leq x_{1}\leq 3.6;\;0.7\leq x_{2}\leq 0.8;\;17\leq x_{3}\leq 28;\;7.3\leq x_{4}\leq 8.3;$$


$$7.3\leq x_{5}\leq 8.3;\;2.9\leq x_{6}\leq 3.9;\;5\leq x_{7}\leq 5.5;$$



The experimental results are presented in Figure 20 and Table 15. As shown in Table 15, the stability of LSEWOA in the Speed Reducer design problem significantly surpasses other algorithms, and it achieves the highest optimization accuracy among all algorithms. This indicates that LSEWOA has a significant advantage in handling such optimization problems.

### 8.4. Cantilever Beam

A cantilever beam is a common structural form, fixed at one end and free at the other, as shown in Figure 21. The cantilever beam design problem is a classic engineering structural optimization problem, with the objective of minimizing material usage or beam weight while satisfying constraints on strength, stability, and other factors. This optimization problem is widely used in civil engineering, mechanical design, and aerospace fields.

The cantilever beam consists of five hollow square cross-section units. As shown in Figure 21, each unit is defined by one variable, and the thickness is constant. Therefore, the design problem includes five structural parameters, which correspond to five decision variables, denoted as $s_{1}$, $s_{2}$, $s_{3}$, $s_{4}$, $s_{5}$. The objective function for the cantilever beam design problem can be expressed as:


*Variable:*

$$\begin{array}{c} x=\left[s_{1},s_{2},s_{3},s_{4},s_{5}\right]=\left[x_{1},x_{2},x_{3},x_{4},x_{5}\right] \end{array}$$




*Minimize:*

(50)
$$f\left(x\right)=0.0624\left(x_{1}+x_{2}+x_{3}+x_{4}+x_{5}\right)$$




*Subject to:*

(51)
$$g\left(x\right)=\frac{61}{x_{1}^{3}}+\frac{37}{x_{2}^{3}}+\frac{19}{x_{3}^{3}}+\frac{7}{x_{4}^{3}}+\frac{1}{x_{5}^{3}}-1\leq 0$$




*Variable range:*

$$\begin{array}{c} 0.01\leq x_{i}\leq 100,\;i=1,2,3,4,5 \end{array}$$



The experimental results are presented in Figure 22 and Table 15. As shown in Table 15, the stability of LSEWOA in the Cantilever Beam design problem significantly surpasses other algorithms, and it achieves the highest optimization accuracy among all algorithms. This indicates that LSEWOA has a significant advantage in handling such optimization problems.

### 8.5. I-Beam

An I-beam, named for its cross-sectional shape resembling the letter ‘I’, is a type of steel with high strength and low self-weight. It is widely used in various engineering structures. Its superior mechanical properties make it applicable in multiple fields, particularly in structures subjected to bending moments and axial forces. The objective of I-beam design optimization is to select the geometric parameters of the I-beam (such as width, height, thickness, etc.) in a way that maximizes its performance. This typically involves maximizing its load-bearing capacity, minimizing material usage, controlling structural deformations, and reducing costs. Optimizing I-beam design in engineering can enhance the safety, economy, and efficiency of structures. As shown in Figure 23, the I-beam design optimization problem involves four variables ($x_{1}$, $x_{2}$, $x_{3}$ and $x_{4}$) and two constraints ($g_{1}$ and $g_{2}$). $x_{1}$, $x_{2}$, $x_{3}$ and $x_{4}$ represent the web height, flange width, web thickness, and flange thickness of the I-beam, respectively. The objective function for the I-beam design problem can be described as:


*Variable:*

x=[x1,x2,x3,x4]




*Maximize:*

(52)
$$f\left(x\right)=\frac{5000}{\frac{x_{3}\cdot {\left(x_{1}-2x_{4}\right)}^{3}}{12}+\frac{x_{2}\cdot x_{4}^{3}}{6}+2x_{2}\cdot x_{4}{\left(\frac{x_{1}-x_{4}}{2}\right)}^{2}}$$




*Subject to:*

(53)
$$g_{1}\left(x\right)=2x_{2}\cdot x_{3}+x_{3}\cdot \left(x_{1}-2x_{4}\right)-300\leq 0;$$


(54)
$$g_{2}\left(x\right)=\frac{18\cdot 10^{4}x_{1}}{x_{3}{\left(x_{1}-2x_{4}\right)}^{3}+2x_{2}\cdot x_{3}\left(4x_{4}^{2}+3x_{1}\cdot \left(x_{1}-2x_{4}\right)\right)}\leq 0$$




*Variable range:*

$$10\leq x_{1}\leq 80;\;10\leq x_{2}\leq 50;\;0.9\leq x_{3}\leq 5;\;0.9\leq x_{4}\leq 5;$$



The experimental results are shown in Figure 24 and Table 15. As shown in Table 15, LSEWOA significantly outperforms the other algorithms in terms of both optimization accuracy and stability for the I-beam design problem. This demonstrates that LSWOA has superior solving capabilities when handling this type of problem. This indicates that LSEWOA has a significant advantage in handling such optimization problems.

### 8.6. Piston Lever

The piston lever is a typical mechanical structure, as shown in Figure 25, and its design problem is a classic engineering optimization problem. It involves the design adjustment of multiple geometric and mechanical parameters, with the goal of minimizing material usage or structural weight while ensuring that constraints such as strength and stability are satisfied. This type of optimization problem is widely applied in mechanical engineering, vehicle design, and other industrial scenarios, particularly when aiming for lightweight and efficient designs of moving components.

In the piston lever optimization problem, the objective is to minimize the total material consumption of the piston lever while ensuring that the structural strength and performance meet the design requirements. The geometric structure of the piston lever is defined by several design parameters, which describe the relationships of its key dimensions. From the geometric relationships, the variables $x_{1}$ to $x_{4}$ can be interpreted as follows: $x_{1}$, $x_{2}$ are the main length and width parameters of the geometric structure, controlling the overall lever arm; $x_{3}$ is the cross-sectional radius at the point of force application, influencing the force distribution; $x_{4}$ is a geometric dimension related to the support point. The objective function for the piston lever design problem can be described as:


*Variable:*

x=[x1,x2,x3,x4]




*Minimize:*

(55)
$$f\left(x\right)=0.25\pi x_{3}^{2}\left(L_{2}-L_{1}\right)$$




*Subject to:*

(56)
$$g_{1}\left(x\right)=QL\cos\theta -RF\leq 0;$$


(57)
$$g_{2}\left(x\right)=Q\left(L-x_{4}\right)-M_{\max}\leq 0;$$


(58)
$$g_{3}\left(x\right)=1.2\left(L_{2}-L_{1}\right)-L_{1}\leq 0;$$


(59)
$$g_{4}\left(x\right)=\frac{x_{3}}{2}-x_{2}\leq 0;$$




*Variable range:*

$$\begin{array}{c} 0.05\leq x_{1}\leq 500;\;0.05\leq x_{2}\leq 500;\;0.05\leq x_{4}\leq 500;\;0.05\leq x_{3}\leq 120; \end{array}$$




*Where:*

$$\begin{array}{c} Q=10000;\;P=1500;\;L=240;\;M_{\max}=1.8\times 10^{6}; \end{array}$$


$$\begin{array}{c} L_{1}=\sqrt{{\left(x_{4}-x_{2}\right)}^{2}+x_{1}^{2}};\;L_{2}=\sqrt{{\left(x_{4}\sin\theta +x_{1}\right)}^{2}+{\left(x_{2}-x_{4}\cos\theta \right)}^{2}}; \end{array}$$


$$\begin{array}{c} R=\frac{|-x_{4}\left(x_{4}\sin\theta +x_{1}\right)+x_{1}\left(x_{2}-x_{4}\cos\theta \right)|}{\sqrt{{\left(x_{4}-x_{2}\right)}^{2}+x_{1}^{2}}}; \end{array}$$


$$\begin{array}{c} F=0.25\pi Px_{3}^{2}; \end{array}$$



The experimental results are shown in Figure 26 and Table 15. From Table 15, it can be seen that in the piston lever design optimization problem, LSEWOA significantly outperforms other algorithms in both optimization accuracy and stability. This indicates that LSEWOA has a significant advantage when dealing with such optimization problems.

### 8.7. Multi-Disc Clutch Brake

The multiple-disc clutch brake is commonly used in transmission systems or mechanical devices to enhance the performance and efficiency of the clutch brake [44]. The structure of the multiple-disc clutch brake is shown in Figure 27. The objective of the multiple-disc clutch brake optimization problem is to minimize the system’s cost by adjusting the design parameters of the clutch brake system (such as the thickness of discs, inner&outer radius, actuating force, and the number of friction surfaces), while satisfying various constraints. The multiple-disc clutch brake optimization problem involves five design variables and eight constraints. The meanings of the variables $x_{1}$ to $x_{5}$: $x_{1}$ indicates the inner radius; $x_{2}$ indicates the outer radius; $x_{3}$ indicates the thickness of discs; $x_{4}$ indicates the actuating force; $x_{5}$ indicates the number of friction surfaces. The objective function for the Multi-disc Clutch Brake design problem can be described as:


*Variable:*

$$\begin{array}{c} x=[x_{1},x_{2},x_{3},x_{4},x_{5}] \end{array}$$




*Minimize:*

(60)
y=f(x)




*Subject to:*

(61)
$$g_{1}=\Delta r+x_{1}-x_{2}\leq 0;$$


(62)
$$g_{2}=\left(x_{5}+1\right)\left(x_{3}+\delta \right)-l_{\max}\leq 0;$$


(63)
$$g_{3}=P_{rz}-P_{max}\leq 0;$$


(64)
$$g_{4}=P_{rz}\cdot V_{sr}-P_{max}\cdot V_{srmax}\leq 0;$$


(65)
$$g_{5}=V_{sr}-V_{srmax}\leq 0;$$


(66)
$$g_{6}=T-T_{max}\leq 0;$$


(67)
$$g_{7}=s\cdot M_{s}-M_{h}\leq 0;$$


(68)
$$g_{8}=-T\leq 0;$$




*Where:*

$$\begin{array}{c} f\left(x\right)=\pi x_{3}\rho \left(x_{2}^{2}-x_{1}^{2}\right)\left(x_{5}+1\right). \end{array}$$


$$\begin{array}{c} M_{h}=\frac{2}{3}\mu Fx_{5}\frac{x_{2}^{3}-x_{1}^{3}}{x_{2}^{2}-x_{1}^{2}}; \end{array}$$


$$\begin{array}{c} P_{rz}=\frac{F}{\pi \left(x_{2}^{2}-x_{1}^{2}\right)}; \end{array}$$


$$\begin{array}{c} V_{sr}=\frac{2\pi n\left(x_{2}^{3}-x_{1}^{3}\right)}{90\left(x_{2}^{2}-x_{1}^{2}\right)}; \end{array}$$


$$\begin{array}{c} T=\frac{I_{z}\pi n}{30\left(M_{h}+M_{f}\right)}; \end{array}$$


$$\begin{array}{c} P_{rz}=\frac{x_{4}}{\pi \cdot (x_{2}^{2}-x_{1}^{2})}; \end{array}$$


$$\begin{array}{c} V_{sr}=\pi \cdot R_{sr}\cdot \frac{n}{30}; \end{array}$$


$$\begin{array}{c} R_{sr}=\frac{2}{3}\cdot \frac{x_{2}^{3}-x_{1}^{3}}{x_{2}^{2}*x_{1}^{2}}; \end{array}$$


$$\begin{array}{c} \Delta r=20;t_{max}=3;t_{min}=1.5;l_{max}=30;Z_{max}=10;V_{max}=10;\mu =0.6;\delta =0.5;M_{s}=40; \end{array}$$


$$\begin{array}{c} M_{f}=3;n=250;P_{max}=1;I_{z}=55;T_{max}=15;Fmax=1000;r_{imin}=55; \end{array}$$


$$\begin{array}{c} r_{omax}=110;\rho =0.0000078; \end{array}$$




*Variable range:*

$$60\leq x_{1}\leq 80;\;90\leq x_{2}\leq 110;\;1\leq x_{3}\leq 3;\;0\leq x_{4}\leq 1000;2\leq x_{5}\leq 9$$



The experimental results are shown in Figure 28 and Table 15. From Table 15, it can be observed that in the multiple-disc clutch brake optimization problem, LSEWOA significantly outperforms other algorithms in both optimization accuracy and stability.

### 8.8. Gas Transmission System

The gas transmission system is a crucial component of the modern energy supply chain, widely used in various industries, urban natural gas supply, and multinational energy transportation. Since the transportation of natural gas relies on Gas Transmission Compressors and pipeline networks, the design optimization of these devices is essential to ensuring energy transmission efficiency and reducing energy waste. The objective of the Gas Transmission Compressor optimization problem is to design and optimize the parameters of the natural gas transmission compressor, so that the compressor can deliver optimal performance under different working conditions, reduce energy consumption, extend service life, and minimize costs. As shown in Figure 29, the Gas Transmission Compressor optimization problem involves four design variables and one constraint. The meanings of the variables $x_{1}$ to $x_{4}$ are: $x_{1}$ indicates the length between compressor stations; $x_{2}$ indicates the compression ratio denoting inlet pressure to the compressor; $x_{3}$ indicates the pipe inside diameter; $x_{4}$ indicates the gas speed on the output side. The mathematical modeling of the Gas Transmission Compressor optimization problem is as follows:


*Variable:*

x=[x1,x2,x3,x4]




*Minimize:*

(69)
$$y=8.61\cdot 10^{5}x_{1}^{\frac{1}{2}}x_{2}x_{3}^{-\frac{2}{3}}x_{4}^{-\frac{1}{2}}+3.69\cdot 10^{4}x_{3}+7.72\cdot 10^{8}x_{1}^{-1}x_{2}^{0.219}-765.43\cdot 10^{6}x_{1}^{-1}$$




*Subject to:*

(70)
$$g=x_{4}x_{2}^{-2}+x_{2}^{-2}-1\leq 0;$$




*Variable range:*

$$20<x_{1}<50;\;1<x_{2}<10;\;20<x_{3}<45;\;0.1<x_{4}<60$$



The experimental results are shown in Figure 30 and Table 15. From Table 15, it can be seen that in the Gas Transmission Compressor optimization problem, LSEWOA significantly outperforms other algorithms in both optimization accuracy and stability.

### 8.9. Industrial Refrigeration System

In the chemical plant design, an industrial refrigeration system is one of the key auxiliary facilities, widely used in chemical production processes, especially in operations such as chemical reactions, storage, transportation, and refining, where temperature control and heat exchange are critical. Chemical plants often require significant cooling and temperature control to maintain reaction stability, ensure product quality, reduce energy consumption and emissions, and ensure the proper functioning of equipment. Therefore, industrial refrigeration systems play a crucial role in the design of chemical plants. The industrial refrigeration system design problem focuses on minimizing energy consumption and cost while ensuring efficient cooling performance, as shown in Figure 31. The objective is to configure the system components, such as compressors, condensers, and evaporators, to achieve the lowest operating cost and optimal heat exchange efficiency. The problem includes fourteen variables: compressor power $x_{1}$ and $x_{2}$, refrigerant flow rate and mass flow $x_{3}$ through $x_{6}$, characteristics of the condenser and evaporator $x_{7}$ and $x_{8}$, compression ratios $x_{9}$ and $x_{10}$, temperature parameters $x_{11}$ and $x_{12}$, and flow rate parameters $x_{13}$ and $x_{14}$. Specifically, compressor power $x_{1}$ and $x_{2}$ control the cooling capacity; refrigerant flow rate and mass flow $x_{3}$ through $x_{6}$ indicate the refrigerant flow through condensers, evaporators, and receivers; $x_{7}$ and $x_{8}$ represent the sizing parameters of the condenser and evaporator; $x_{9}$ and $x_{10}$ define the compression degree and compressor efficiency; $x_{11}$ and $x_{12}$ manage the temperature differential for heat exchange; and $x_{13}$ and $x_{14}$ govern the flow rate of cooling water or refrigerant, affecting overall system performance. Industrial refrigeration system design problem is modeled below.


*Variable:*

$$\begin{array}{cc} x & =[x_{1},x_{2},x_{3},x_{4},x_{5},x_{6},x_{7},x_{8},x_{9},x_{10},x_{11},x_{12},x_{13},x_{14}] \end{array}$$




*Minimize:*

(71)
y=f(x)




*Subject to:*

(72)
$$g_{1}=\frac{1.524}{x_{7}}-1\leq 0;$$


(73)
$$g_{2}=\frac{1.524}{x_{8}}-1\leq 0;$$


(74)
$$g_{3}=0.07789\cdot x_{1}-\frac{2\cdot x_{9}}{x_{7}}-1\leq 0;$$


(75)
$$g_{4}=\frac{7.05305\cdot x_{1}^{2}\cdot x_{10}}{x_{9}\cdot x_{8}\cdot x_{2}\cdot x_{14}}-1\leq 0;$$


(76)
$$g_{5}=\frac{0.0833\cdot x_{14}}{x_{13}}-1\leq 0;$$


(77)
$$g_{6}=\frac{47.136\cdot x_{2}^{0.333}\cdot x_{12}}{x_{10}}-1.333\cdot x_{8}\cdot x_{13}^{2.1195}+\frac{62.08\cdot x_{13}^{2.1195}\cdot x_{8}^{0.2}}{x_{12}\cdot x_{10}}-1\leq 0;$$


(78)
$$g_{7}=0.04771\cdot x_{10}\cdot x_{8}^{1.8812}\cdot x_{12}^{0.3424}-1\leq 0;$$


(79)
$$g_{8}=0.0488\cdot x_{9}\cdot x_{7}^{1.893}\cdot x_{11}^{0.316}-1\leq 0;$$


(80)
$$g_{9}=\frac{0.0099\cdot x_{1}}{x_{3}}-1\leq 0;$$


(81)
$$g_{10}=\frac{0.0193\cdot x_{2}}{x_{4}}-1\leq 0;$$


(82)
$$g_{11}=\frac{0.0298\cdot x_{1}}{x_{5}}-1\leq 0;$$


(83)
$$g_{12}=\frac{0.056\cdot x_{2}}{x_{6}}-1\leq 0;$$


(84)
$$g_{13}=\frac{2}{x_{9}}-1\leq 0;$$


(85)
$$g_{14}=\frac{2}{x_{10}}-1\leq 0;$$


(86)
$$g_{15}=\frac{x_{12}}{x_{11}}-1\leq 0;$$




*Where:*

$$f\left(x\right)=63098.88\cdot x_{2}\cdot x_{4}\cdot x_{12}+5441.5\cdot x_{2}^{2}\cdot x_{12}+115055.5\cdot x_{2}^{1.664}\cdot x_{6}$$


$$+6172.27\cdot x_{2}^{2}\cdot x_{6}+63098.88\cdot x_{1}\cdot x_{3}\cdot x_{11}+5441.5\cdot x_{1}^{2}\cdot x_{11}$$


$$+115055.5\cdot x_{1}^{1.664}\cdot x_{5}+6172.27\cdot x_{1}^{2}\cdot x_{5}+140.53\cdot x_{1}\cdot x_{11}$$


$$+281.29\cdot x_{3}\cdot x_{11}+70.26\cdot x_{1}^{2}+281.29\cdot x_{1}\cdot x_{3}+281.29\cdot x_{3}^{2}$$


$$+14437\cdot x_{8}^{1.8812}\cdot x_{12}^{0.3424}\cdot x_{10}\cdot x_{1}^{2}\cdot \frac{x_{7}}{x_{14}\cdot x_{9}}$$


$$+20470.2\cdot x_{7}^{2.893}\cdot x_{11}^{0.316}\cdot x_{12}$$




*Variable range:*

$$0.001<x_{1}<5;\;0.001<x_{2}<5;\;0.001<x_{3}<5;\;0.001<x_{4}<5;$$


$$0.001<x_{5}<5;\;0.001<x_{6}<5;\;0.001<x_{7}<5;\;0.001<x_{8}<5;$$


$$0.001<x_{9}<5;\;0.001<x_{10}<5;\;0.001<x_{11}<5;\;0.001<x_{12}<5;$$


$$0.001<x_{13}<5;\;0.001<x_{14}<5;$$



The experimental results, presented in Figure 32 and Table 15. The results demonstrate that LSEWOA consistently escapes local optima, continuously searching for better solutions even when other algorithms are trapped in sub-optimal states. Compared with other algorithms, LSWOA shows exceptional stability and accuracy in solution-seeking. This indicates that LSEWOA has a significant advantage when dealing with such optimization problems.

## 9. Conclusions

The Whale Optimization Algorithm (WOA) suffers from several issues, including premature convergence, low population diversity, slow convergence speed, low convergence accuracy, and an imbalance between exploration and exploitation. These drawbacks make WOA less competitive in solving complex, real-world optimization problems. To address these limitations, this paper presents an enhanced WOA, LSEWOA, which integrates multiple strategies aimed at better balancing exploration and exploitation, improving convergence speed, and enhancing optimization accuracy.

In ablation study, we validated the significance of five improvement strategies incorporated into LSEWOA. A qualitative analysis experiment were conducted to examine the search behavior of LSEWOA across different functions, the ratio of exploitation to exploration, and population diversity. The results demonstrated that LSEWOA effectively explored the solution space and identified either the optimal or near-optimal solutions. The exploitation-exploration ratio charts show that LSEWOA successfully balanced exploration and exploitation. The population diversity curve indicates that LSEWOA maintained high population diversity, avoiding premature convergence, and continued until it approaches an optimal solution.

In comparasin experiments, LSEWOA was tested on 23 selected classic benchmark functions alongside superior WOA variants and other state-of-the-art metaheuristic algorithms in 30/50/100 dimensions. The results after 30 runs show that LSEWOA was highly competitive, achieving optimal or near-optimal solutions with higher efficiency. In nine engineering design optimization problems, LSEWOA demonstrated strong optimization capability and stability, indicating that LSEWOA can be used as an optimization tool for addressing complex real-world problems. LSEWOA provides new insights into the application of WOA in real-world scenarios.

In future research, we will conduct further studies by rigorously testing using prototypes of various mechanical components, validating against real-world scenarios, and incorporating practical constraints into the optimization process to achieve more reliable and effective mechanical design. Ultimately, LSEWOA aims to improve the reliability and effectiveness of engineering design, aligning with contemporary industrial demands. We recommend LSEWOA as a tool for design, simulation, and manufacturing, for use by researchers and practitioners in the field. In the future, we will also explore more application scenarios of LSEWOA and extend its use to more challenging benchmark functions, such as path planning, multi-objective problems, constrained optimization problems, and parameter optimization.

## Figures and Tables

**Figure 1 sensors-25-02054-f001:**
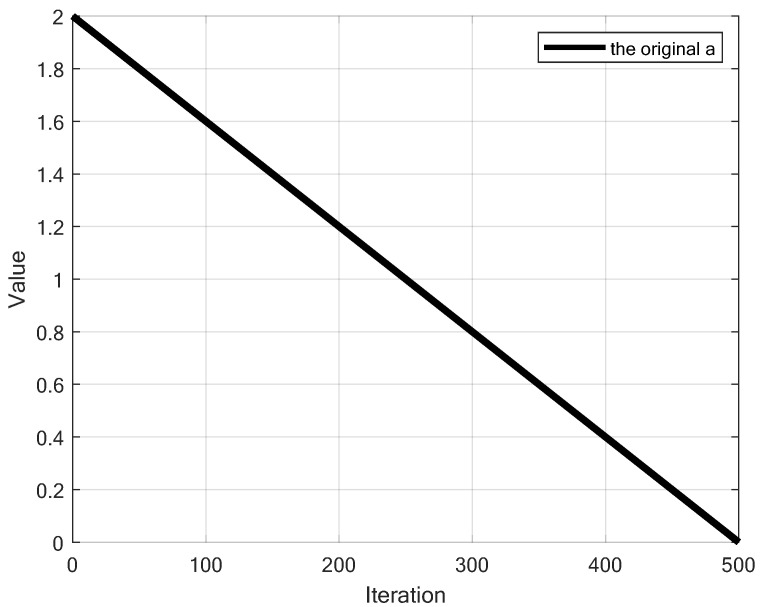
The linearly decreasing convergence factor *a*.

**Figure 2 sensors-25-02054-f002:**
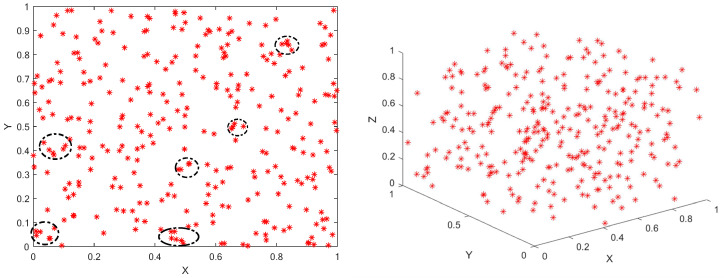
Population initialized by pseudo-random number (*N* = 300). The portions enclosed by the dashed black line represent the phenomenon of population aggregation.

**Figure 3 sensors-25-02054-f003:**
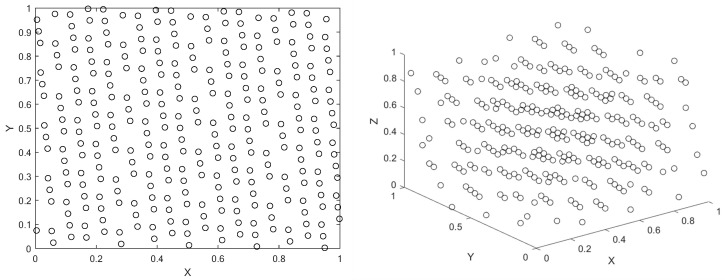
Population initialized by Good Nodes Set (*N* = 300).

**Figure 4 sensors-25-02054-f004:**
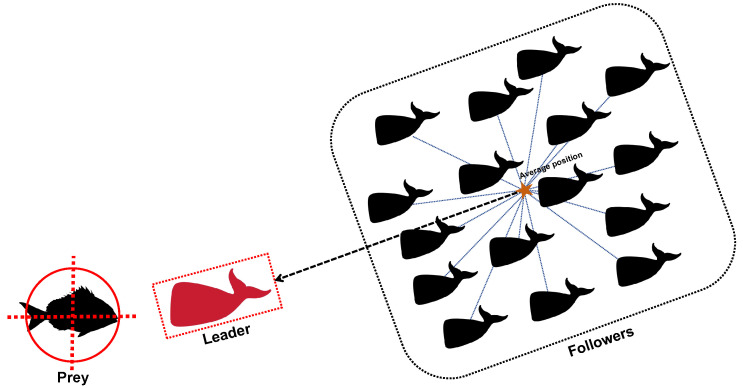
Leader-Followers Search-for-Prey Strategy.

**Figure 5 sensors-25-02054-f005:**
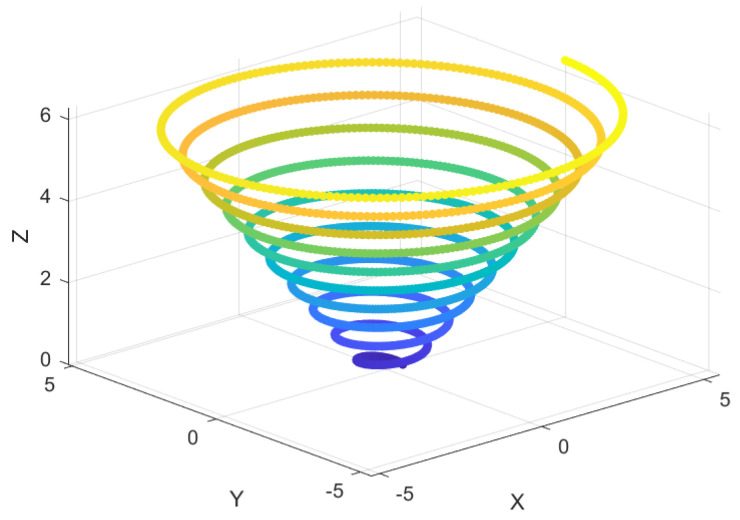
Simulation of Spiral flight.

**Figure 6 sensors-25-02054-f006:**
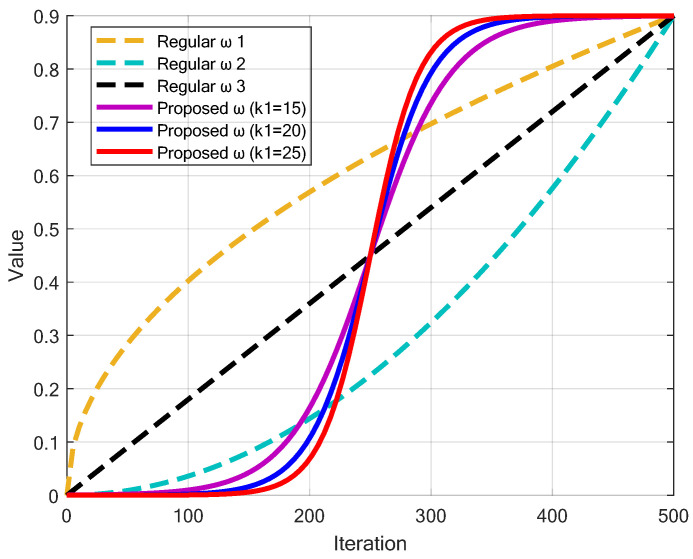
Comparison of different types of inertia weight ω. Regular ω 1: $\omega =0.9\cdot \frac{t}{T}$; Regular ω 2: $\omega =0.9\cdot {\left(\frac{t}{T}\right)}^{2}$; Regular ω 1: $\omega =0.9\cdot \sqrt{\frac{t}{T}}$.

**Figure 7 sensors-25-02054-f007:**
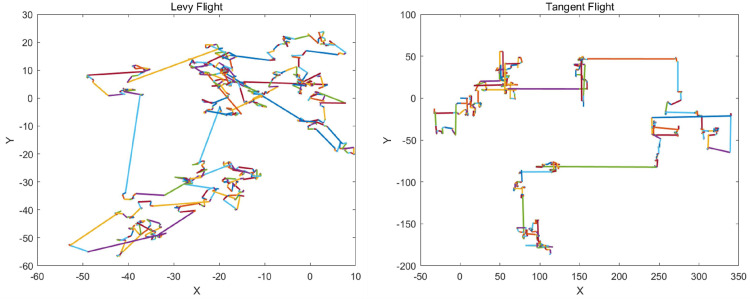
Comparison of Levy flight and Tangent flight.

**Figure 8 sensors-25-02054-f008:**
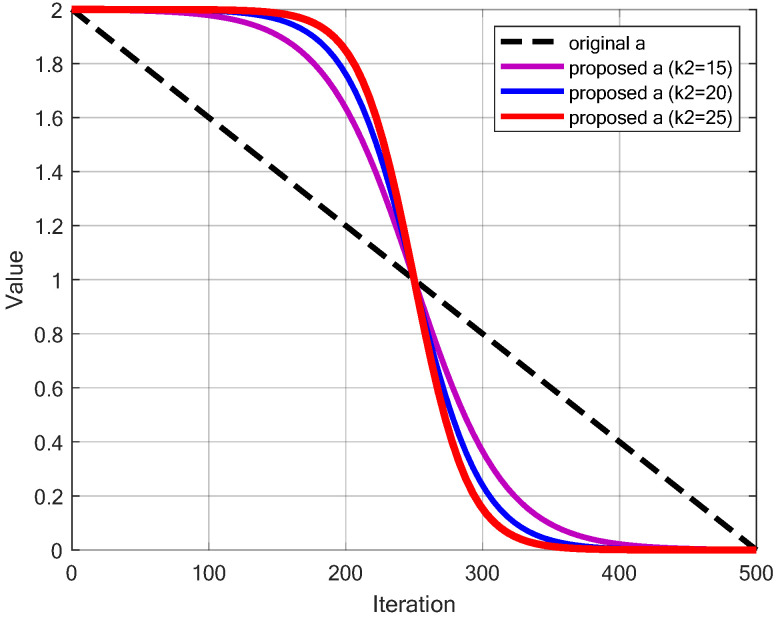
Comparison of the original convergence factor *a* and the proposed *a*.

**Figure 9 sensors-25-02054-f009:**
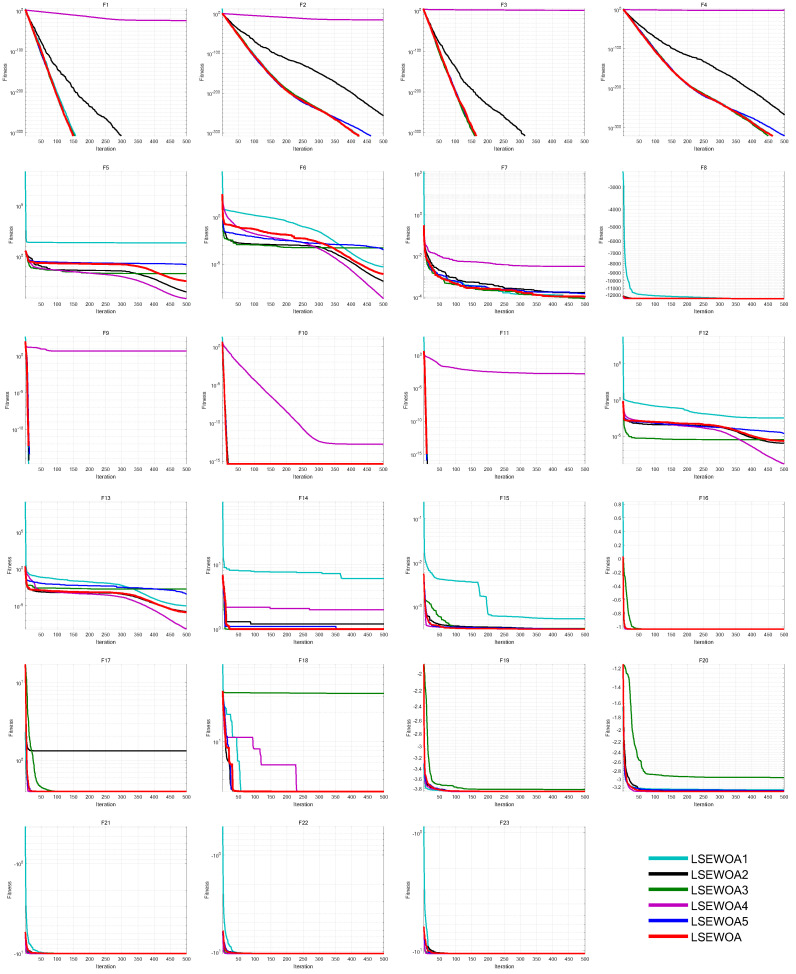
Iteration curves of the LSEWOAs in ablation study.

**Figure 10 sensors-25-02054-f010:**
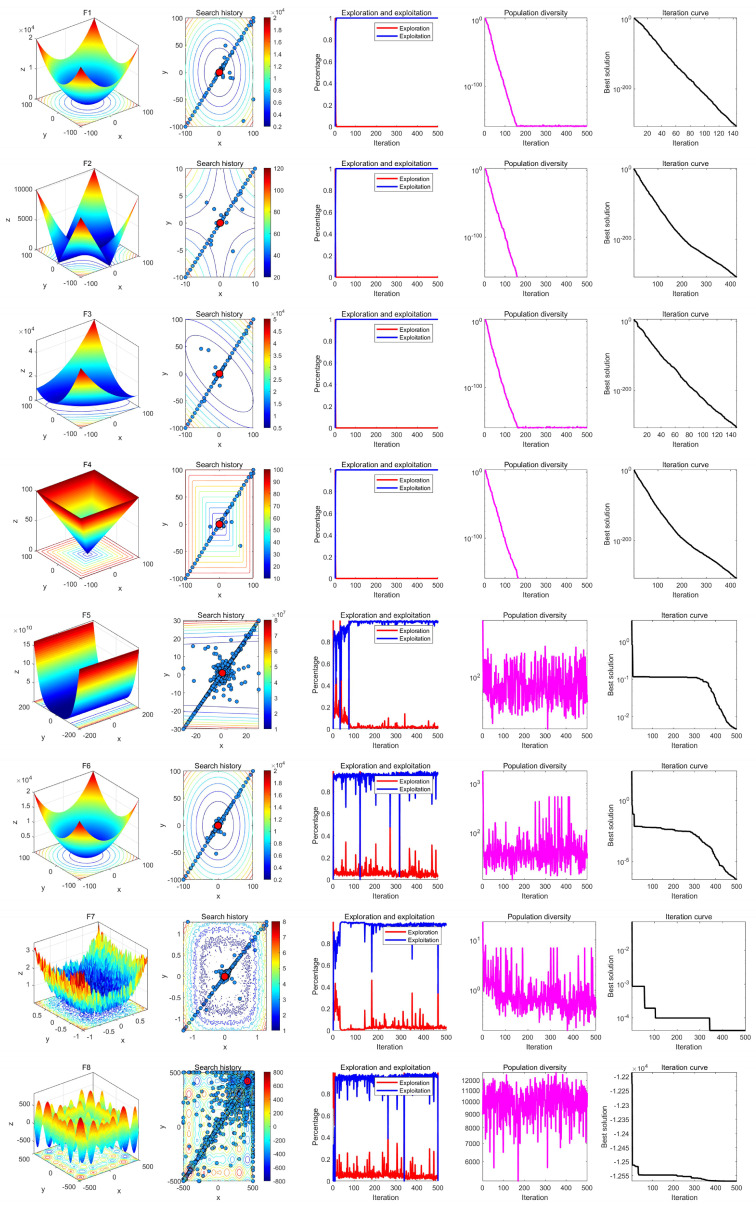
Results of qualitative analysis experiment (F1–F8).

**Figure 11 sensors-25-02054-f011:**
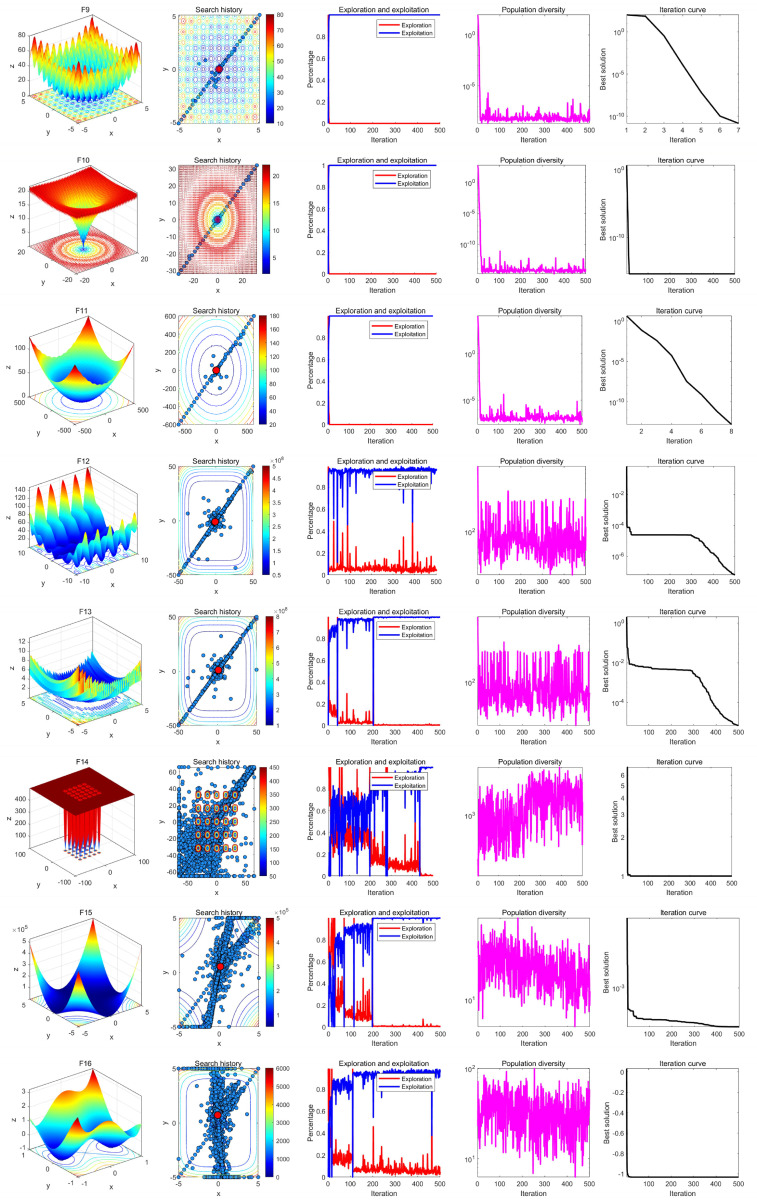
Results of qualitative analysis experiment (F9–F16).

**Figure 12 sensors-25-02054-f012:**
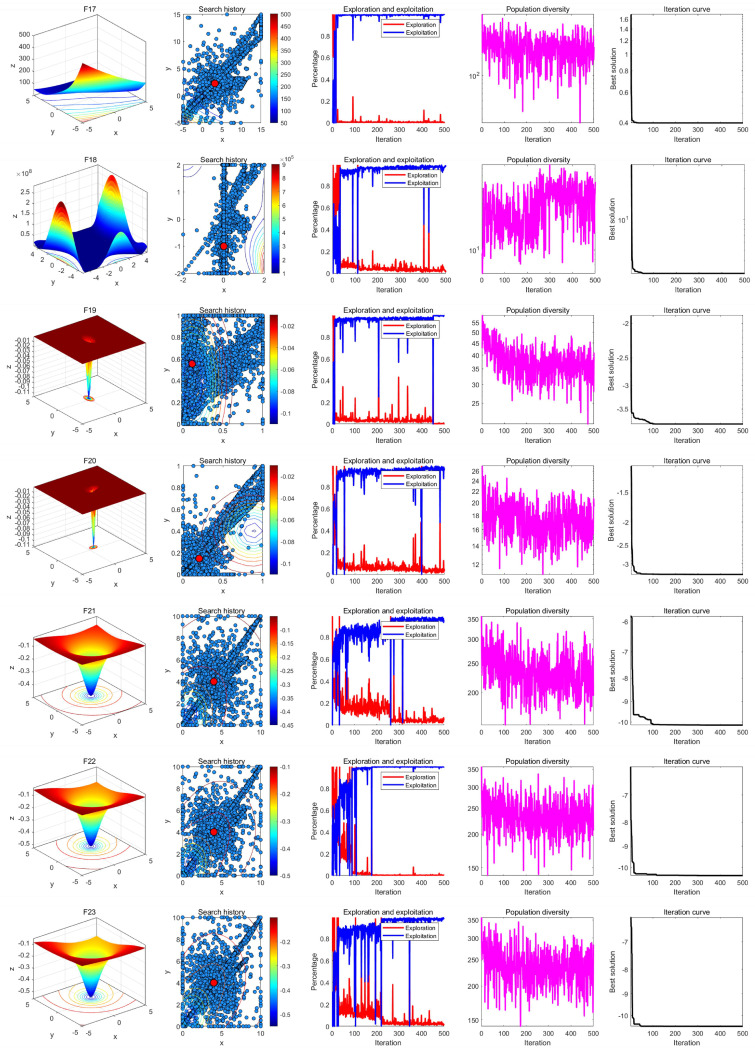
Results of qualitative analysis experiment (F17–F23).

**Figure 13 sensors-25-02054-f013:**
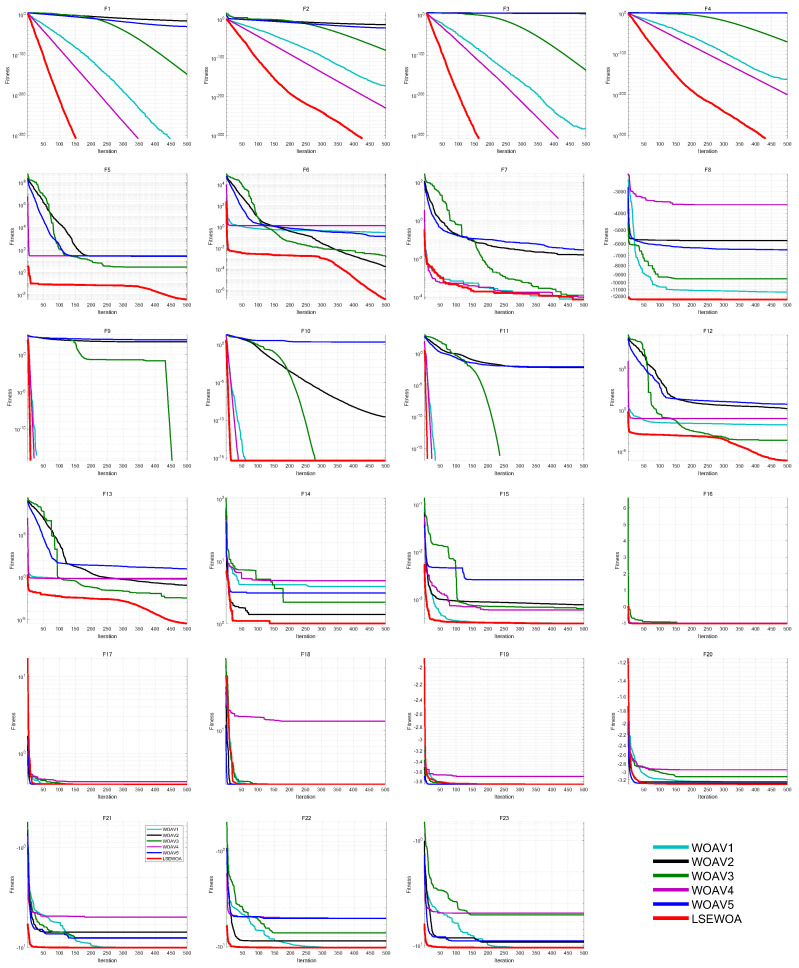
Iterative curves of different WOA variants in the comparison experiment in 30 dimensions.

**Figure 14 sensors-25-02054-f014:**
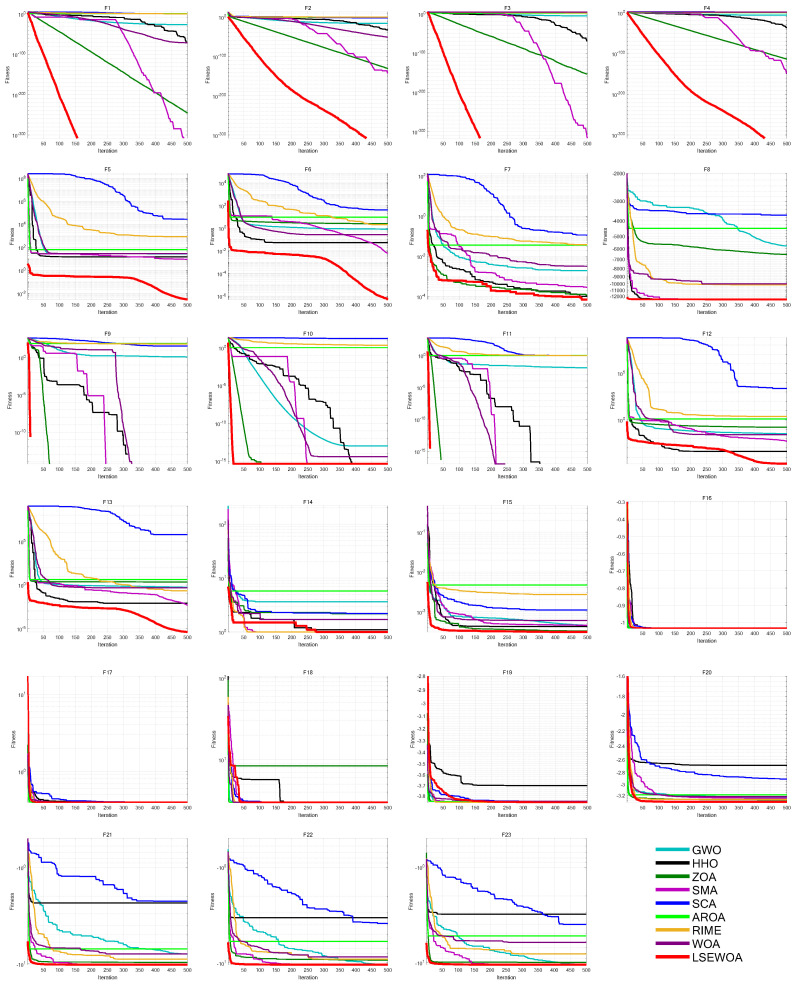
Iterative curves of different algorithms in comparison experiment in 30 dimensions.

**Figure 15 sensors-25-02054-f015:**
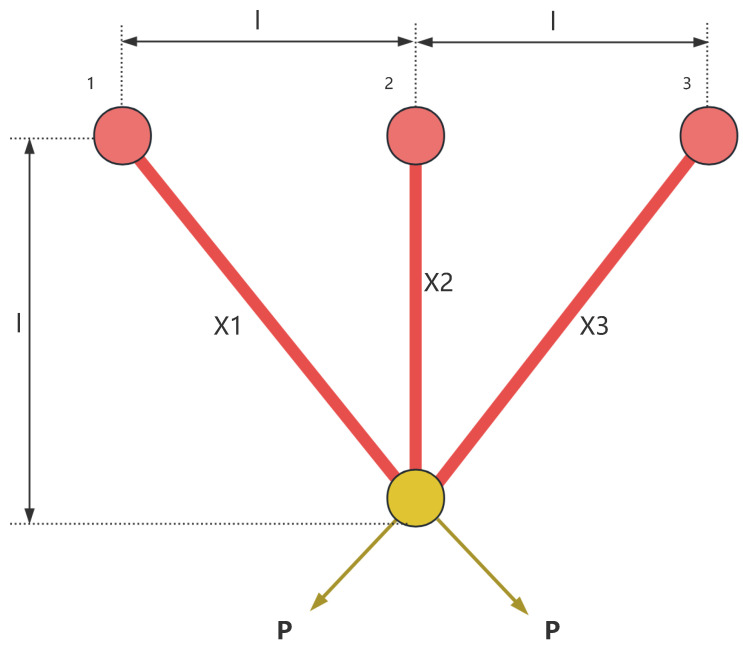
The structure of a three-bar truss.

**Figure 16 sensors-25-02054-f016:**
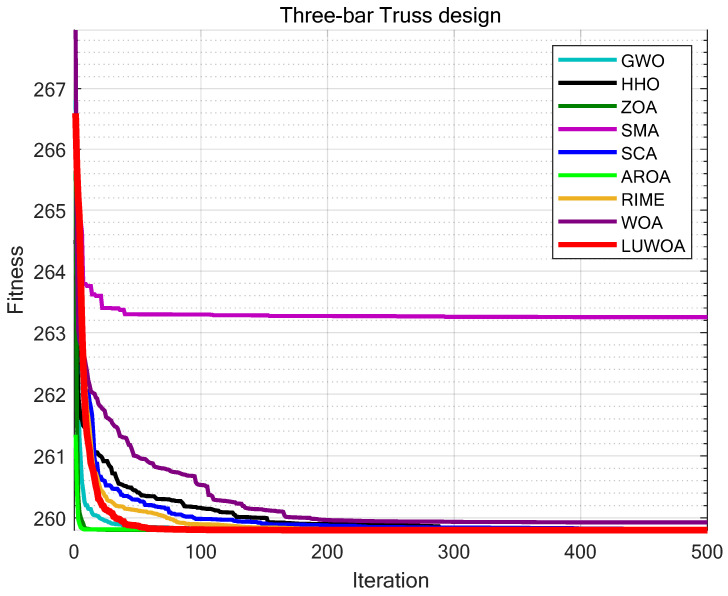
Iteration curves of the algorithms in Three-bar Truss design.

**Figure 17 sensors-25-02054-f017:**
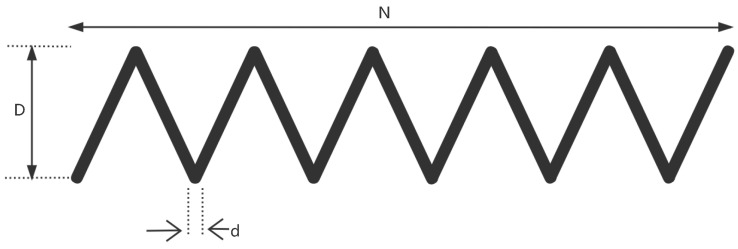
The structure of a tension/compression spring.

**Figure 18 sensors-25-02054-f018:**
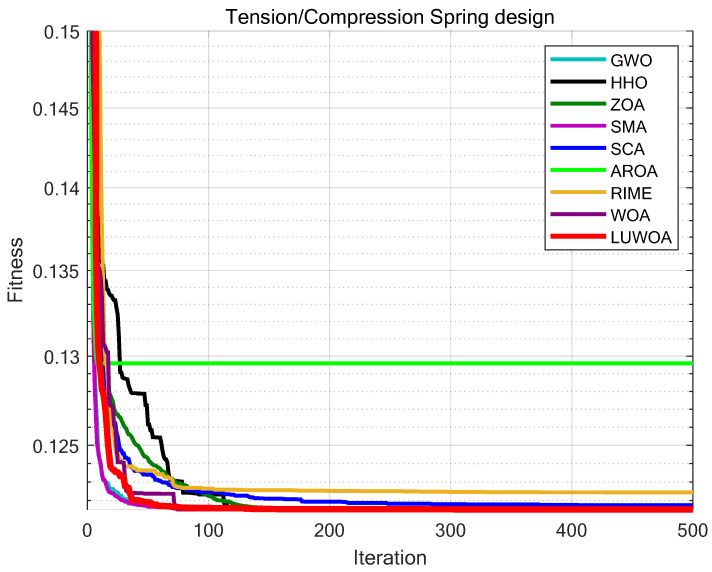
Iteration curves of the algorithms in Tension/Compression Spring design.

**Figure 19 sensors-25-02054-f019:**
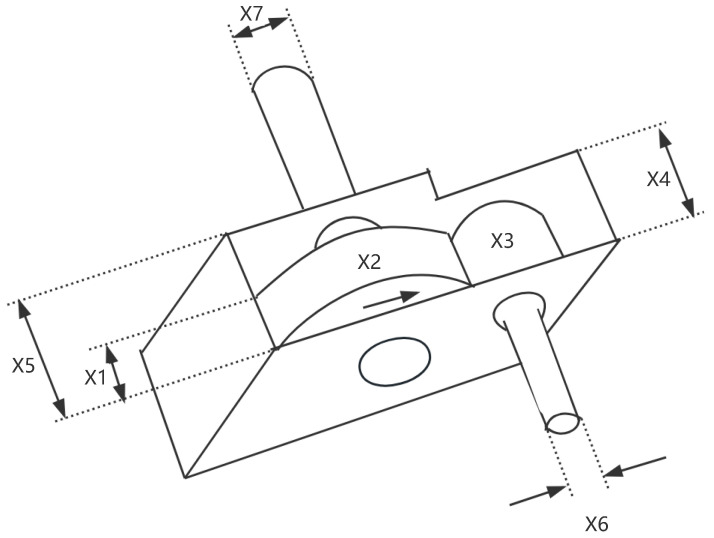
The structure of a speed reducer.

**Figure 20 sensors-25-02054-f020:**
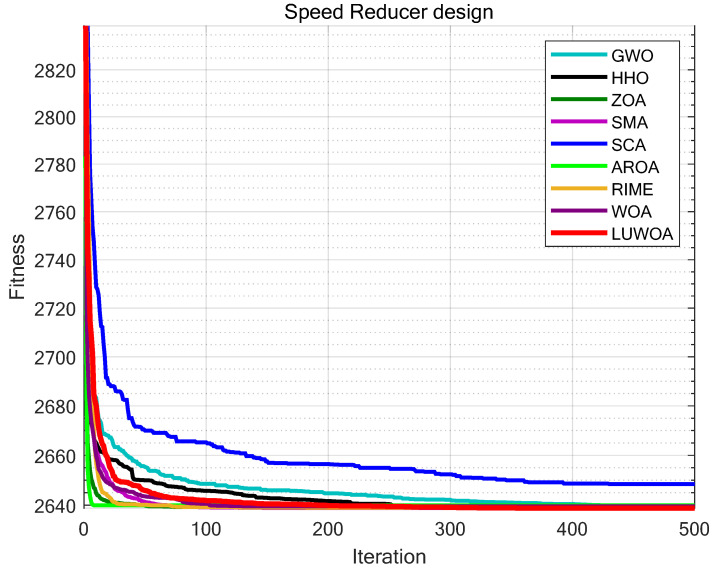
Iteration curves of the algorithms in Speed Reducer design.

**Figure 21 sensors-25-02054-f021:**
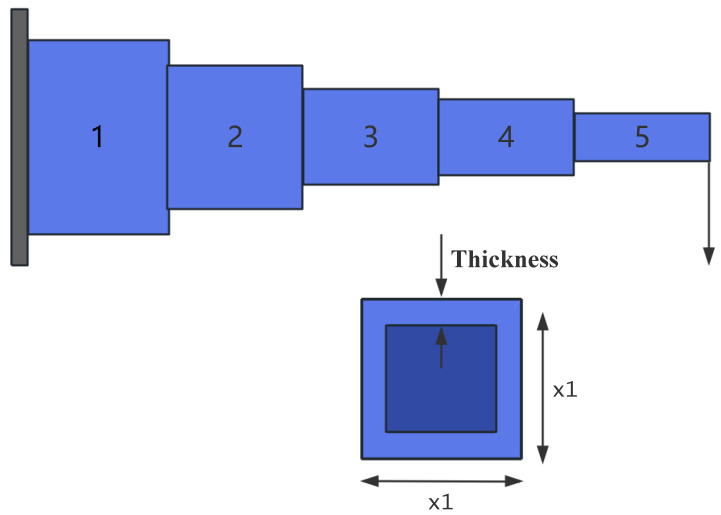
The structure of a cantilever beam.

**Figure 22 sensors-25-02054-f022:**
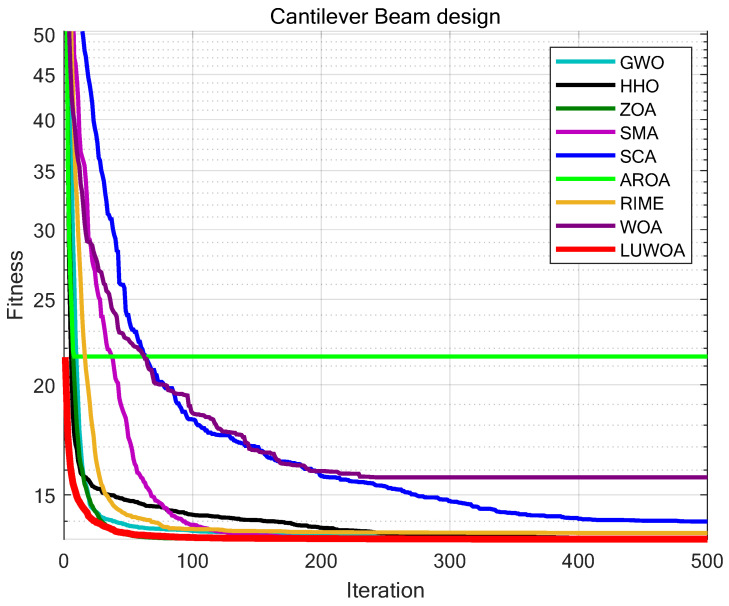
Iteration curves of the algorithms in Cantilever Beam design.

**Figure 23 sensors-25-02054-f023:**
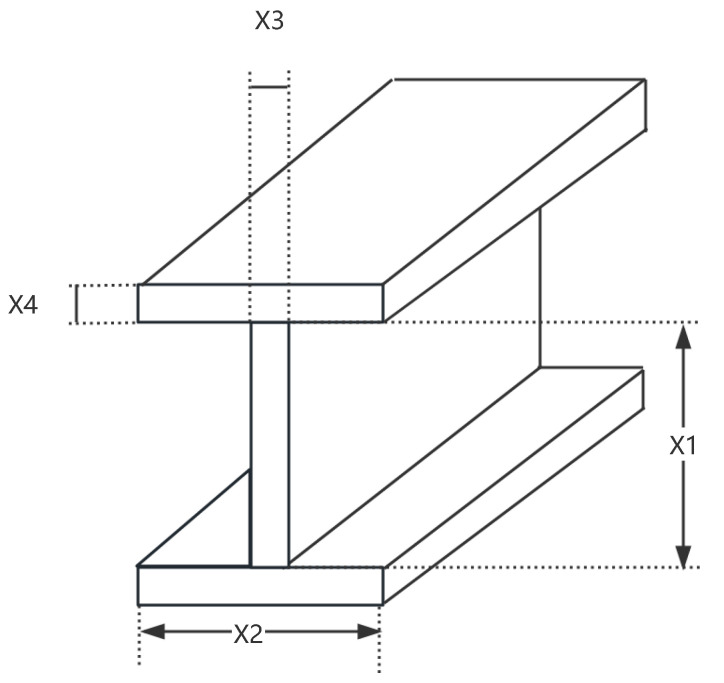
The structure of an I-beam.

**Figure 24 sensors-25-02054-f024:**
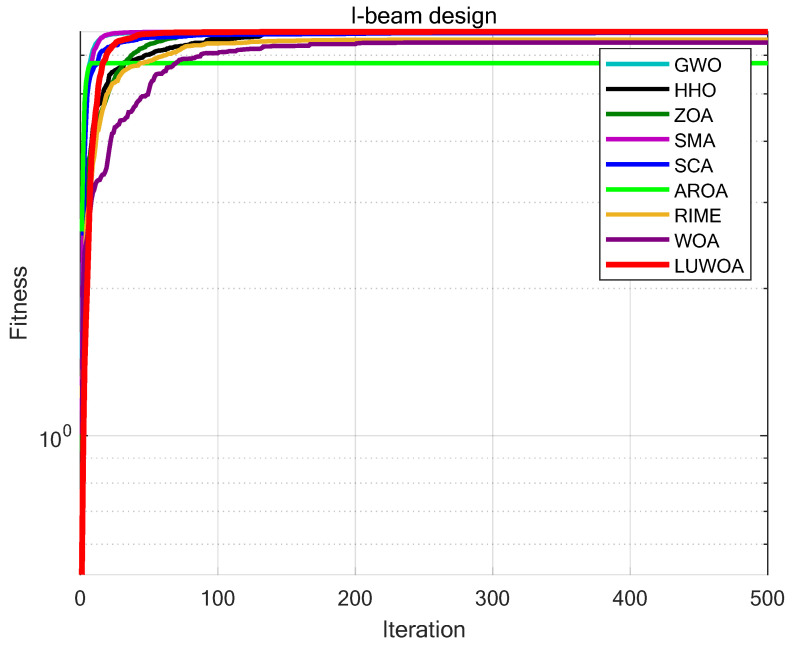
Iteration curves of the algorithms in I-beam design.

**Figure 25 sensors-25-02054-f025:**
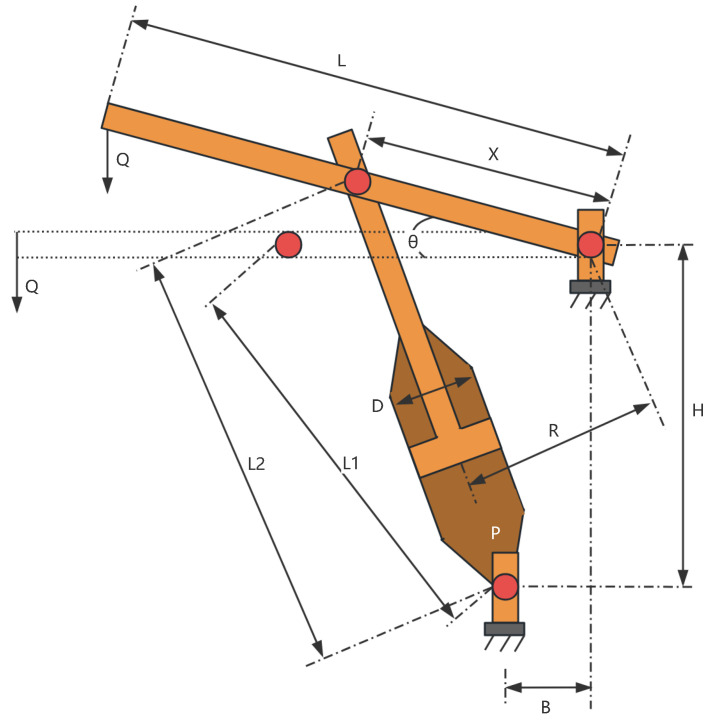
The structure of a piston lever.

**Figure 26 sensors-25-02054-f026:**
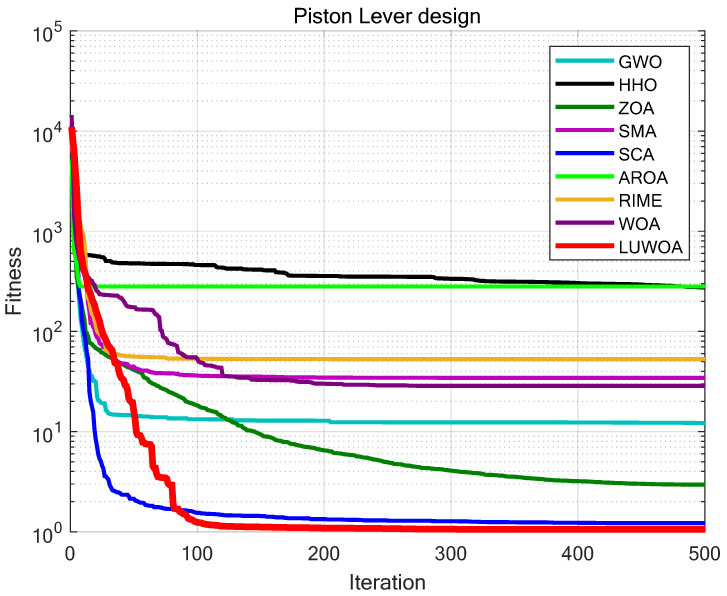
Iteration curves of the algorithms in Piston Lever design.

**Figure 27 sensors-25-02054-f027:**
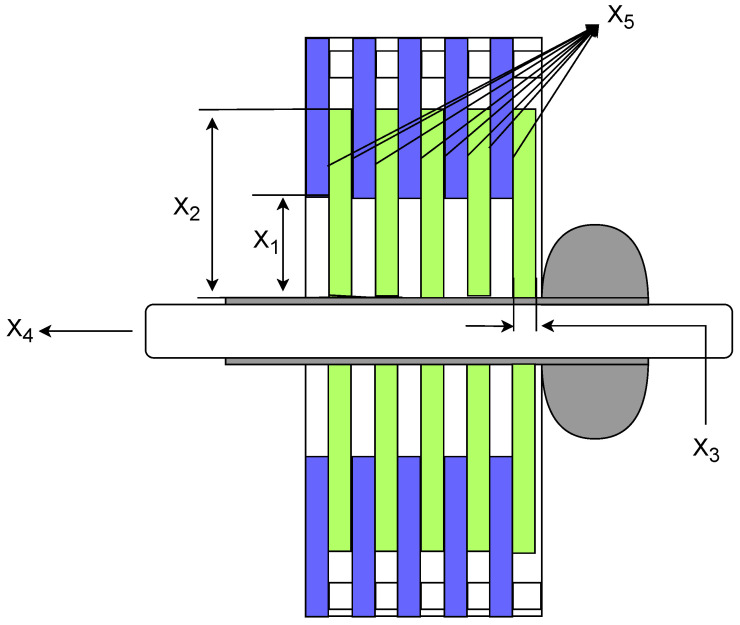
The structure of a multiple-disc clutch brake.

**Figure 28 sensors-25-02054-f028:**
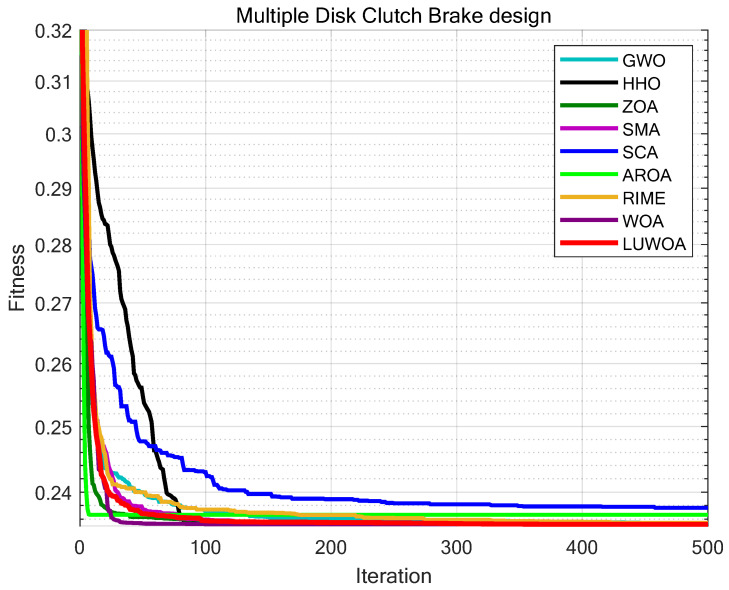
Iteration curves of the algorithms in Multiple-disc Clutch Brake design.

**Figure 29 sensors-25-02054-f029:**
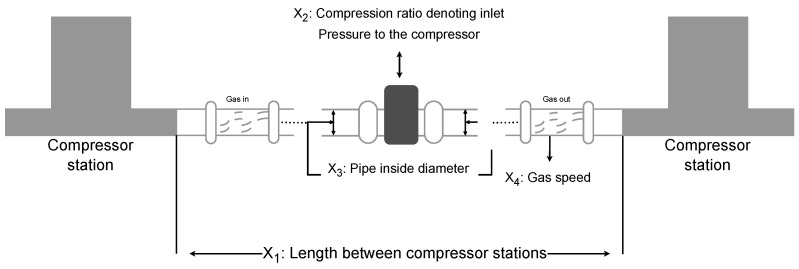
The structure of a gas transmission system.

**Figure 30 sensors-25-02054-f030:**
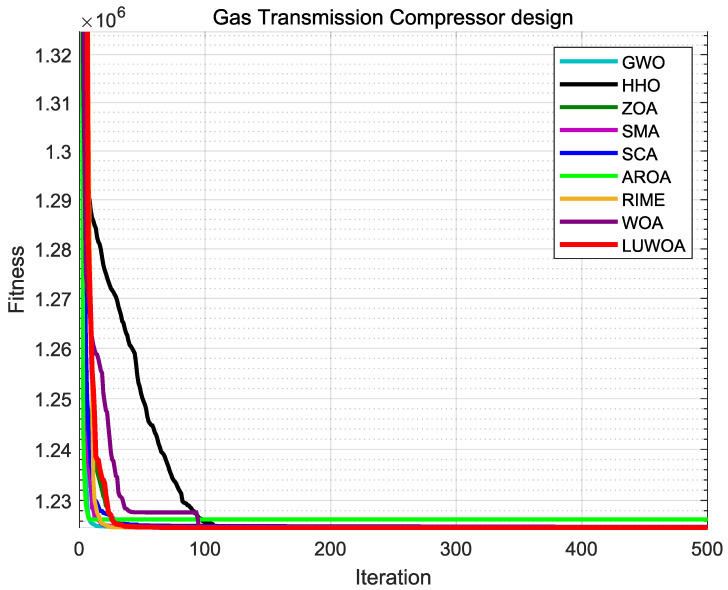
Iteration curves of the algorithms in Gas Transmission System design.

**Figure 31 sensors-25-02054-f031:**
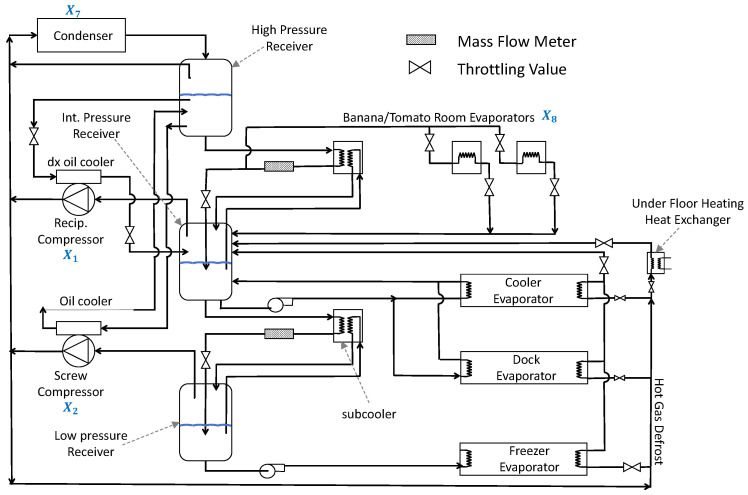
The structure of an industrial refrigeration system.

**Figure 32 sensors-25-02054-f032:**
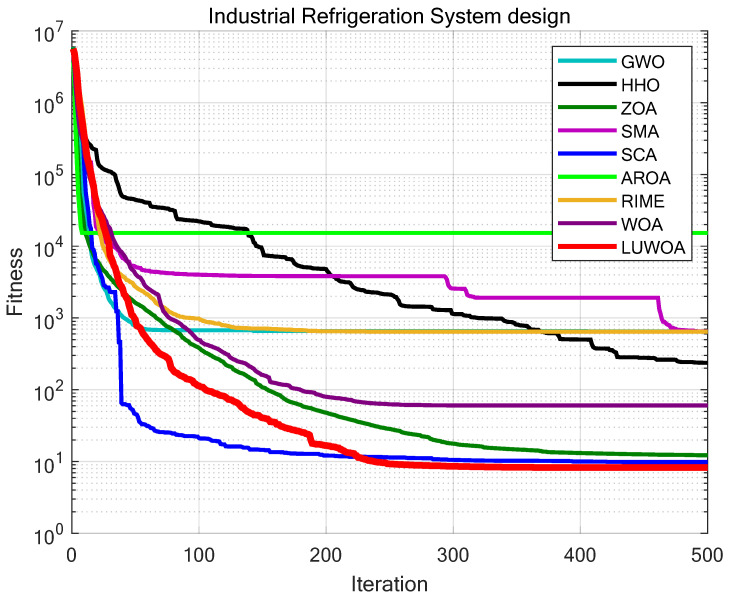
Iteration curves of the algorithms in Industrial Refrigeration System design.

**Table 1 sensors-25-02054-t001:** Basic metaheuristic algorithms.

Algorithm	Year	Author	Source of Inspiration
Simulated annealing (SA) [1]	1953	Metropolis et al.	The annealing process.
Genetic Algorithm (GA) [2]	1975	John Holland et al.	Darwin’s theory of evolution and Mendel’s genetic.
Ant Colony Optimization (ACO) [3]	1991	Dorigo et al.	Foraging behavior of ants.
Particle Swarm Optimization (PSO) [4]	1995	Kennedy et al.	Foraging behavior of birds.
Differential Evolution (DE) [5]	1997	Rainer Storn et al.	Mutation, crossover and selection.
Artificial Bee Colony (ABC) [6]	2005	Karaboga et al.	Honeybee’s foraging behavior.
Harris Hawks Optimization (HHO) [7]	2019	Elhamifar et al.	Hunting behavior of Harris hawks.
Aquila Optimizer (AO) [8]	2021	Laith Abualigah et al.	Hunting behavior of aquila eagles.
Beluga Whale Optimization (BWO) [9]	2022	C Zhong et al.	Swimming, foraging and whale fall phenomena of beluga white whales.
Crayfish Optimization (COA) [10]	2023	H Jia et al.	foraging, cooling and competitive behaviors of crayfish.

**Table 3 sensors-25-02054-t003:** Standard Benchmark Functions [31].

Function	Function’s Name	Type	Dimension (Dim)	Best Value
F1	Sphere	Uni-modal, Scalable	30/50/100	0
F2	Schwefel’s Problem 2.22	Uni-modal, Scalable	30/50/100	0
F3	Schwefel’s Problem 1.2	Uni-modal, Scalable	30/50/100	0
F4	Schwefel’s Problem 2.21	Uni-modal, Scalable	30/50/100	0
F5	Generalized Rosenbrock’s Function	Uni-modal, Scalable	30/50/100	0
F6	Step Function	Uni-modal, Scalable	30/50/100	0
F7	Quartic Function	Uni-modal, Scalable	30/50/100	0
F8	Generalized Schwefel’s Function	Multi-modal, Scalable	30/50/100	−418.98·Dim
F9	Generalized Rastrigin’s Function	Multi-modal, Scalable	30/50/100	0
F10	Ackley’s Function	Multi-modal, Scalable	30/50/100	0
F11	Generalized Griewank’s Function	Multi-modal, Scalable	30/50/100	0
F12	Generalized Penalized Function 1	Multi-modal, Scalable	30/50/100	0
F13	Generalized Penalized Function 2	Multi-modal, Scalable	30/50/100	0
F14	Shekel’s Foxholes Function	Multi-modal, Unscalable	2	0.998
F15	Kowalik’s Function	Composition, Unscalable	4	0.0003075
F16	Six-Hump Camel-Back Function	Composition, Unscalable	2	−1.0316
F17	Branin Function	Composition, Unscalable	2	0.398
F18	Goldstein-Price Function	Composition, Unscalable	2	3
F19	Hartman’s Function 1	Composition, Unscalable	3	−3.8628
F20	Hartman’s Function 2	Composition, Unscalable	6	−3.32
F21	Shekel’s Function 1	Composition, Unscalable	4	−10.1532
F22	Shekel’s Function 2	Composition, Unscalable	4	−10.4029
F23	Shekel’s Function 3	Composition, Unscalable	4	−10.5364

**Table 4 sensors-25-02054-t004:** Results of parameter sensitivity analysis experiment.

Function	LSEWOA(15, 15)	LSEWOA(15, 20)	LSEWOA(15, 25)	LSEWOA(20, 15)	LSEWOA(20, 20)	LSEWOA(20, 25)	LSEWOA(25, 15)	LSEWOA(25, 20)	LSEWOA(25, 25)
F1	5.0000	5.0000	5.0000	5.0000	5.0000	5.0000	5.0000	5.0000	5.0000
F2	5.1500	5.0600	4.9700	4.9700	4.9700	4.9700	4.9700	4.9700	4.9700
F3	5.0000	5.0000	5.0000	5.0000	5.0000	5.0000	5.0000	5.0000	5.0000
F4	5.6200	5.0100	4.9100	4.9100	4.9100	4.9100	4.9100	4.9100	4.9100
F5	7.0200	4.9600	3.1200	6.7200	4.1000	3.7800	7.1000	4.3200	3.8800
F6	7.9200	4.5600	3.1000	7.2800	4.6600	2.7000	7.7400	4.2400	2.8000
F7	4.4000	4.6800	4.8800	5.1400	6.2200	4.6000	4.9800	5.1400	4.9600
F8	7.9800	5.0000	2.3000	7.9800	4.8200	2.0800	7.7600	4.8800	2.2000
F9	5.0000	5.0000	5.0000	5.0000	5.0000	5.0000	5.0000	5.0000	5.0000
F10	5.0000	5.0000	5.0000	5.0000	5.0000	5.0000	5.0000	5.0000	5.0000
F11	5.0000	5.0000	5.0000	5.0000	5.0000	5.0000	5.0000	5.0000	5.0000
F12	6.4600	3.8800	4.3000	6.6400	4.0400	4.4000	6.6800	3.8200	4.7800
F13	7.1600	3.9800	4.2400	7.0600	4.2600	3.6600	6.8200	3.9000	3.9200
F14	7.6400	4.5600	2.9500	7.5900	5.1300	2.5800	7.4200	4.7900	2.3400
F15	4.7000	4.3800	5.5400	4.4200	4.8800	5.7200	4.8600	4.9400	5.5600
F16	7.6000	5.0900	2.1100	8.0000	5.2000	2.2000	7.8400	4.8100	2.1500
F17	8.1200	5.2400	2.2600	7.7800	5.0000	2.0700	7.7000	4.9200	1.9100
F18	4.2800	4.3000	5.9600	4.0000	5.4000	5.2000	4.2800	5.4800	6.1000
F19	5.8400	4.4800	4.9000	6.4000	4.0800	4.9800	6.0400	4.0200	4.2600
F20	7.6600	5.2600	7.8600	7.8800	4.8400	2.2200	2.2400	2.2600	4.7800
F21	7.9800	4.8800	1.7400	8.1400	5.1000	2.2000	7.8600	5.0400	2.0600
F22	8.1800	4.9800	2.1400	7.8200	5.1600	1.8600	7.9400	4.9200	2.0000
F23	8.1000	4.7600	1.8200	7.8400	5.1000	2.1200	8.0600	5.0600	2.1400
Average	6.3830	4.7852	4.0913	6.3291	4.9074	**3.7935**	6.0957	4.6704	3.9443
Rank	9	5	3	8	6	**1**	7	4	2

**Table 5 sensors-25-02054-t005:** Comparative results of different WOA variants in 30 dimensions.

Function	Metrics	WOAV1	WOAV2	WOAV3	WOAV4	WOAV5	LSEWOA
F1	Ave	0.0000E+00	9.8638E-17	1.3408E-149	0.0000E+00	5.6634E-30	0.0000E+00
	Std	0.0000E+00	2.5718E-16	3.8808E-149	0.0000E+00	1.3704E-29	0.0000E+00
F2	Ave	2.0330E-169	2.1140E-14	4.1665E-81	1.9041E-229	4.4123E-23	0.0000E+00
	Std	2.6530E-169	2.2621E-14	9.4838E-81	2.0640E-229	5.8699E-23	0.0000E+00
F3	Ave	9.7597E-283	1.1229E+04	2.2191E-138	0.0000E+00	1.1728E+03	0.0000E+00
	Std	9.9857E-283	4.4476E+03	8.0227E-138	0.0000E+00	8.1830E+02	0.0000E+00
F4	Ave	1.2894E-165	2.7553E+01	8.5275E-71	1.7281E-200	3.4477E+01	0.0000E+00
	Std	1.6534E-165	1.4118E+01	6.5641E-71	1.8576E-200	1.2159E+01	0.0000E+00
F5	Ave	2.8434E+01	2.5671E+01	3.9129E+00	2.8674E+01	2.5411E+01	7.0226E-04
	Std	1.9274E-01	9.7131E-01	9.8399E+00	1.5929E-01	1.6234E+00	7.1473E-04
F6	Ave	3.0652E-01	1.0685E-04	1.0465E-03	1.5137E+00	1.8336E-02	4.4627E-07
	Std	1.5010E-01	6.0965E-05	9.1790E-04	4.0534E-01	6.2416E-02	8.5647E-07
F7	Ave	1.2704E-04	1.8777E-02	1.6854E-04	1.0146E-04	3.6796E-02	9.0617E-05
	Std	1.6348E-04	8.9501E-03	1.1422E-04	1.0148E-04	2.7993E-02	9.8754E-05
F8	Ave	−1.1587E+04	−5799.7097	−9.6835E+03	−5071.082	−6.2135E+03	−1.2569E+04
	Std	8.6513E+02	1.7944E+02	1.4073E+03	1.9181E+03	3.0826E+02	8.8190E-03
F9	Ave	0.0000E+00	4.3725E+01	7.2948E-02	0.0000E+00	1.0214E+02	0.0000E+00
	Std	0.0000E+00	3.5049E+01	3.9955E-01	0.0000E+00	3.3930E+01	0.0000E+00
F10	Ave	4.4409E-16	5.5139E-10	4.4409E-16	4.4409E-16	3.4122E+00	4.4409E-16
	Std	0.0000E+00	7.5953E-10	0.0000E+00	0.0000E+00	4.4409E-16	0.0000E+00
F11	Ave	0.0000E+00	6.5879E-03	0.0000E+00	0.0000E+00	9.8103E-03	0.0000E+00
	Std	0.0000E+00	1.7301E-02	0.0000E+00	0.0000E+00	1.6744E-02	0.0000E+00
F12	Ave	2.1045E-02	1.5641E+00	2.0810E-02	8.2819E-02	4.5985E+00	1.1056E-06
	Std	1.0780E-02	1.6340E+00	1.1291E-01	3.0729E-02	4.2231E+00	3.3784E-06
F13	Ave	4.0665E-01	8.1110E-02	7.6209E-03	7.0082E-01	7.7391E+00	1.1074E-03
	Std	1.9499E-01	8.3807E-02	1.9818E-02	2.2757E-01	1.2779E+01	3.3682E-03
F14	Ave	4.7805E+00	1.3287E+00	1.5967E+00	6.6164E+00	3.4841E+00	9.9800E-01
	Std	4.4631E+00	7.5207E-01	1.3094E+00	4.6380E+00	3.5220E+00	1.5423E-16
F15	Ave	3.2035E-04	5.6925E-04	1.6779E-03	6.0438E-04	2.4352E-03	3.1131E-04
	Std	4.057E-05	3.4712E-04	3.6112E-03	2.2205E-04	6.0863E-03	8.9892E-06
F16	Ave	−1.0316E+00	−1.0316E+00	−1.0316E+00	−9.9542E-01	−1.0316E+00	−1.0316E+00
	Std	6.3208E-16	6.7122E-16	1.5322E-05	3.5053E-02	6.3208E-16	6.1358E-16
F17	Ave	3.9789E-01	3.9789E-01	3.9807E-01	4.1714E-01	3.9789E-01	3.9789E-01
	Std	1.8233E-09	0.0000E+00	2.4150E-04	2.1294E-02	0.0000E+00	3.0227E-14
F18	Ave	3.0000E+00	3.0000E+00	3.0003E+00	9.7182E+00	3.9000E+00	3.0000E+00
	Std	1.4523E-14	1.5003E-15	2.6184E-04	1.0779E+01	4.9295E+00	9.1567E-05
F19	Ave	−3.8628E+00	−3.8628E+00	−3.8610E+00	−3.7703E+00	−3.8628E+00	−3.8628E+00
	Std	1.6154E-12	2.7101E-15	1.4594E-03	8.7085E-02	2.5684E-15	4.5466E-05
F20	Ave	−3.2970E+00	−3.2705E+00	3.1149E+00	−2.8895E+00	−3.2546E+00	−3.3139E+00
	Std	5.0399E-02	5.9923E-02	2.3645E-02	2.2151E-01	5.9923E-02	1.0705E-02
F21	Ave	−1.0153E+01	−8.0347E+00	−8.3530E+00	−4.7618E+00	−7.1336E+00	−1.0153E+01
	Std	1.2439E-03	2.6741E+00	2.3778E+00	1.0865E+00	3.3901E+00	9.4998E-11
F22	Ave	−1.0402E+01	−8.3325E+00	−6.9258E+00	−4.7869E+00	−6.9124E+00	−1.0403E+01
	Std	4.1015E-03	2.8050E+00	3.5457E+00	8.6837E-01	3.4437E+00	1.0354E-10
F23	Ave	−1.0536E+01	−9.0041E+00	−8.3377E+00	−4.7300E+00	−6.2289E+00	−1.0536E+01
	Std	1.2232E-04	2.6270E+00	3.4851E+00	8.9833E-01	2.9646E+00	1.5291E-10

**Table 6 sensors-25-02054-t006:** Results of non-parametric tests of different WOA variants in 30 dimensions.

Algorithm	Rank	Average Friedman Value	+/=/−
WOAV1	2	3.2188	19/4/0
WOAV2	4	3.8783	23/0/0
WOAV3	3	3.8370	21/2/0
WOAV4	6	4.3645	18/5/0
WOAV5	2	4.0304	23/0/0
LSEWOA	1	1.6710	−

**Table 7 sensors-25-02054-t007:** Effectiveness of LSEWOA and other WOA variants.

Metrics	WOAV1 (*w*/*t*/*l*)	WOAV2 (*w*/*t*/*l*)	WOAV3 (*w*/*t*/*l*)	WOAV4 (*w*/*t*/*l*)	WOAV5 (*w*/*t*/*l*)	LSEWOA (*w*/*t*/*l*)
*D* = 30	0/4/19	0/0/23	0/2/21	0/5/18	0/0/23	18/5/0
*D* = 50	1/4/18	0/0/23	1/2/20	1/5/17	0/0/23	17/5/1
*D* = 100	1/4/18	0/0/23	1/2/20	1/5/17	0/0/23	17/5/1
Total	2/12/55	0/0/69	2/6/61	2/15/52	0/0/69	52/15/2
OE	20.29%	0.00%	11.59%	24.64%	0.00%	**97.10%**

**Table 8 sensors-25-02054-t008:** Results of non-parametric tests of different WOA variants in higher dimensions.

Dimension	Algorithm	Rank	Average Friedman Value	+/=/−
*D* = 50	WOAV1	2	3.1717	22/0/1
	WOAV2	5	4.0130	19/3/1
	WOAV3	3	3.7203	20/3/0
	WOAV4	6	4.3406	19/3/1
	WOAV5	4	3.8754	22/0/1
	LSEWOA	**1**	**1.7790**	−
*D* = 100	WOAV1	6	3.1717	22/0/1
	WOAV2	4	4.0130	19/3/1
	WOAV3	3	3.7203	20/3/0
	WOAV4	2	4.3406	19/3/1
	WOAV5	9	3.8754	22/0/1
	LSEWOA	**1**	**1.5551**	−

**Table 9 sensors-25-02054-t009:** Details of the metaheuristic algorithms.

Algorithm	Year	Author(s)	Source of Inspiration
Grey Wolf Optimizer (GWO) [38]	2014	Seyedali Mirjalili et al.	The leadership hierarchy and
			hunting system of gray wolves.
Harris Hawk Optimization	2019	AA Heidari et al.	The predatory behavior
algorithm (HHO) [7]			of Harris’s hawks.
Zebra Optimization Algorithm	2022	E Trojovská et al.	Foraging and Defense Strategy
(ZOA) [39]			of zebras.
Slime Mould Algorithm (SMA) [40]	2020	S Li et al.	Foraging behavior of slime molds.
Sine Cosine Algorithm	2022	Seyedali Mirjalili	Mathematical model of the
(SCA) [41]			tangent function.
Attraction-Repulsion Optimization	2024	K Cymerys	Attraction-repulsion phenomenon.
Algorithm (AROA) [42]			
Rime optimization algorithm	2023	Su Hang et al.	The formation process of rime
(RIME) [43]			in nature.
Whale Optimization Algorithm	2016	Seyedali Mirjalili et al.	The hunting behavior of
(WOA) [18]			humpback whales.

**Table 10 sensors-25-02054-t010:** Parameter settings for different metaheuristic algorithms.

Algorithm	Parameter	Value
GWO	Convergence factor *a*	2 decrease to 0
HHO	Threshold	0.5
ZOA	*R*	0.1
SMA	*z*	0.03
	$v_{c}$	1 decrease to 0
SCA	*a*	2
AROA	Attraction factor *c*	0.95
	Local search scaling factor 1	0.15
	Local search scaling factor 2	0.6
	Attraction probability 1	0.2
	Local search probability	0.8
	Expansion factor	0.4
	Local search threshold 1	0.9
	Local search threshold 2	0.85
	Local search threshold 3	0.9
RIME	ω	5
WOA	Convergence factor *a*	2 decrease to 0
	Spiral factor *b*	1
LSEWOA	Convergence factor *a*	2 decrease to 0
	Inertia weight ω	0 increase to 0.9
	Spiral factor *k*	1

**Table 11 sensors-25-02054-t011:** Comparative results of different metaheuristic algorithms in 30 dimensions.

Function	Metrics	GWO	HHO	ZOA	SMA	SCA	AROA	RIME	WOA	LSEWOA
F1	Ave	2.0895E-27	1.6901E-56	1.3535E-249	3.4272E-319	1.1493E+01	3.9851E+00	2.1360E+00	3.122E-72	0.0000E+00
	Std	3.4562E-27	9.2573E-56	1.4325E-249	3.653E-319	1.2787E+01	2.7759E+00	1.2179E+00	1.4327E-71	0.0000E+00
F2	Ave	9.8045E-17	7.2027E-38	2.4225E-130	6.1088E-142	1.5330E-02	7.2023E-01	1.5665E+00	1.0548E-49	0.0000E+00
	Std	6.078E-17	2.6592E-37	8.6167E-130	3.3458E-141	2.0414E-02	2.2962E-01	1.0821E+00	4.055E-49	0.0000E+00
F3	Ave	2.6528E-05	6.4586E-66	2.317E-154	9.577E-296	7.5000E+03	1.9846E+02	1.4642E+03	4.7902E+04	0.0000E+00
	Std	1.1192E-04	3.5282E-65	1.2691E-153	9.5432E-296	5.2636E+03	2.7161E+02	4.5706E+02	1.6365E+04	0.0000E+00
F4	Ave	6.8653E-07	1.2801E-36	4.223E-114	3.4892E-159	3.7443E+01	1.8233E+00	7.5008E+00	5.4785E+01	0.0000E+00
	Std	7.816E-07	5.0091E-36	1.4688E-113	1.9017E-158	1.2096E+01	8.0749E-01	3.2426E+00	2.5634E+01	0.0000E+00
F5	Ave	2.7214E+01	1.2597E+01	2.8435E+01	7.9363E+00	8.2844E+04	9.4260E+01	3.8156E+02	2.7943E+01	2.1804E-04
	Std	7.6933E-01	1.4341E+01	4.6934E-01	1.1387E+01	1.4598E+05	6.3976E+01	5.9200E+02	4.7547E-01	1.9583E-04
F6	Ave	7.5490E-01	1.1604E-01	2.6854E+00	5.6277E-03	2.3240E+01	1.0211E+01	2.0228E+00	3.7597E-01	2.4235E-07
	Std	4.1382E-01	2.0970E-01	5.4786E-01	2.7533E-03	3.3771E+01	3.0057E+00	6.5657E-01	1.9799E-01	3.0661E-07
F7	Ave	2.1070E-03	1.6577E-04	1.2183E-04	2.0192E-04	1.3017E-01	3.3554E-02	4.2646E-02	3.3823E-03	9.2310E-05
	Std	1.1727E-03	2.0006E-04	9.8147E-05	1.4495E-04	1.2503E-01	2.6544E-02	1.7281E-02	3.4709E-03	9.4816E-05
F8	Ave	−6.0646E+03	−12569.413	−6.5366E+03	−12569.0803	−3.7756E+03	−4.5711E+03	−9.9880E+03	−1.0387E+04	−12569.4810
	Std	8.6178E+02	1.8809E-01	6.6667E+02	2.4234E-01	2.9546E+02	7.0335E+02	4.8839E+02	1.9207E+03	7.1526E-03
F9	Ave	2.5289E+00	0.0000E+00	0.0000E+00	0.0000E+00	3.6167E+01	5.1770E+01	6.7959E+01	1.8948E-15	0.0000E+00
	Std	4.8781E+00	0.0000E+00	0.0000E+00	0.0000E+00	3.2372E+01	6.5822E+01	1.2253E+01	1.0378E-14	0.0000E+00
F10	Ave	1.0501E-13	4.4409E-16	4.4409E-16	4.4409E-16	1.2727E+01	8.2919E-01	2.1549E+00	3.76E-15	4.4409E-16
	Std	2.0391E-14	0.0000E+00	0.0000E+00	0.0000E+00	9.4703E+00	3.8480E-01	5.0321E-01	2.6279E-15	0.0000E+00
F11	Ave	4.0717E-03	0.0000E+00	0.0000E+00	0.0000E+00	8.5792E-01	9.8457E-01	9.9393E-01	3.7007E-18	0.0000E+00
	Std	9.2179E-03	0.0000E+00	0.0000E+00	0.0000E+00	2.4644E-01	1.1340E-01	3.4509E-02	2.027e-17	0.0000E+00
F12	Ave	4.5354E-02	7.6756E-04	1.6897E-01	8.3303E-03	1.3194E+04	1.3013E+00	3.3013E+00	2.5013E-02	1.1132E-06
	Std	2.4764E-02	1.6439E-03	6.6607E-02	9.6578E-03	4.3933E+04	3.2708E-01	2.1232E+00	2.5786E-02	1.8246E-06
F13	Ave	6.1504E-01	4.8366E-02	2.2662E+00	6.2369E-03	1.2142E+05	4.0630E+00	2.1907E-01	4.5703E-01	7.3709E-04
	Std	2.3088E-01	9.7235E-02	2.7059E-01	6.4921E-03	2.2536E+05	4.7363E-01	6.5521E-02	2.0359E-01	2.7942E-03
F14	Ave	4.2279E+00	1.4941E+00	2.7431E+00	9.9800E-01	1.6626E+00	6.4242E+00	9.9800E-01	1.9840E+00	9.9800E-01
	Std	4.2406E+00	8.5423E-01	2.0626E+00	1.1416E-12	9.4904E-01	4.4124E+00	6.8753E-12	2.0261E+00	4.2751E-16
F15	Ave	4.4172E-03	3.8813E-04	1.7074E-03	5.4518E-04	9.5536E-04	4.2943E-03	7.1893E-03	7.4320E-04	3.0960E-04
	Std	8.1125E-03	7.6667E-05	5.0750E-03	2.4870E-04	3.4759E-04	6.0518E-03	1.2544E-02	5.6744E-04	5.1968E-06
F16	Ave	−1.0316E+00	−1.0316E+00	−1.0316E+00	−1.0316E+00	−1.0316E+00	−1.0316E+00	−1.0316E+00	−1.0316E+00	−1.0316E+00
	Std	2.5294E-08	1.1537E-06	3.6246E-10	1.0767E-09	6.1383E-05	2.8188E-05	9.1271E-08	2.3332E-09	1.19E-15
F17	Ave	3.9791E-01	3.9817E-01	3.9789E-01	3.9789E-01	3.9913E-01	3.9889E-01	3.9789E-01	3.9791E-01	3.9789E-01
	Std	1.0285E-04	1.0967E-03	2.5577E-08	8.5519E-08	9.2873E-04	3.1123E-03	7.8052E-07	3.9929E-05	1.8276E-14
F18	Ave	3.0001E+00	3.9007E+00	5.7000E+00	3.0000E+00	3.0001E+00	3.9922E+00	3.0000E+00	3.9004E+00	3.0000E+00
	Std	9.6496E-05	4.9298E+00	8.2385E+00	1.0303E-10	1.6082E-04	4.9382E+00	6.9447E-07	4.9312E+00	5.6509E-05
F19	Ave	−3.8615E+00	−3.8024E+00	−3.8623E+00	−3.8628E+00	−3.8549E+00	−3.8591E+00	−3.8628E+00	−3.8535E+00	−3.8628E+00
	Std	2.5697E-03	7.5654E-02	4.5151E-04	1.9366E-07	3.3445E-03	5.6257E-03	2.5256E-07	1.5083E-02	4.59E-05
F20	Ave	−3.2599E+00	−2.5371E+00	−3.3178E+00	−3.2583E+00	−2.9681E+00	−3.2171E+00	−3.2665E+00	−3.2337E+00	3.3220E+00
	Std	7.5667E-02	4.3035E-01	2.1897E-02	6.0626E-02	2.8675E-01	7.5154E-02	6.0327E-02	1.2029E-01	2.5098E-12
F21	Ave	−9.3930E+00	−2.9686E+00	−9.8132E+00	−1.0153E+01	−2.3603E+00	−6.5719E+00	−7.0419E+00	−7.8461E+00	−1.0153E+01
	Std	2.0038E+00	1.4372E+00	1.2934E+00	3.6940E-04	1.9595E+00	3.2936E+00	3.0763E+00	2.6970E+00	6.4992E-11
F22	Ave	−1.0401E+01	−3.6079E+00	−9.6935E+00	−1.0403E+01	−3.0380E+00	−6.9924E+00	−8.8560E+00	−7.5651E+00	−1.0403E+01
	Std	1.3707E-03	1.1968E+00	1.8374E+00	3.3640E-04	1.5471E+00	3.3121E+00	2.9167E+00	2.8153E+00	8.9896E-11
F23	Ave	−1.0264E+01	−3.0811E+00	−9.6350E+00	−1.0536E+01	−3.5309E+00	−6.0164E+00	−9.0563E+00	−6.6449E+00	−1.0536E+01
	Std	1.4812E+00	1.4661E+00	2.0499E+00	2.5217E-04	2.1461E+00	3.3571E+00	2.7900E+00	3.6057E+00	1.1943E-10

**Table 12 sensors-25-02054-t012:** Results of non-parametric tests of different metaheuristic algorithms in 30 dimensions.

Algorithm	Rank	Average Friedman Value	+/=/−
GWO	5	5.2312	23/0/0
HHO	4	4.9297	20/3/0
ZOA	3	3.8956	20/3/0
SMA	2	2.7565	20/3/0
SCA	9	8.0783	23/0/0
AROA	8	7.2913	23/0/0
RIME	7	6.0000	22/0/1
WOA	6	5.2587	23/0/0
LSEWOA	1	1.5587	−

**Table 13 sensors-25-02054-t013:** Effectiveness of LSEWOA and other metaheuristic algorithms.

Metrics	GWO (*w*/*t*/*l*)	HHO (*w*/*t*/*l*)	ZOA (*w*/*t*/*l*)	SMA (*w*/*t*/*l*)	SCA (*w*/*t*/*l*)	AROA (*w*/*t*/*l*)	RIME (*w*/*t*/*l*)	WOA (*w*/*t*/*l*)	LSEWOA (*w*/*t*/*l*)
*D* = 30	0/0/23	0/3/20	0/3/20	0/3/20	0/0/23	0/0/23	1/0/22	0/0/23	22/1/0
*D* = 50	1/0/22	1/3/19	0/3/20	1/3/19	1/0/22	0/0/23	1/0/22	1/1/21	19/3/1
*D* = 100	1/0/22	1/3/19	0/3/20	1/3/19	1/0/22	0/0/23	1/0/22	1/2/20	19/3/1
Total	2/0/67	2/9/58	0/9/60	2/9/58	2/0/67	0/0/69	3/0/66	2/3/64	60/7/2
OE	2.90%	15.94%	13.04%	15.94%	2.90%	0.00%	4.35%	7.25%	**97.10%**

**Table 14 sensors-25-02054-t014:** Results of non-parametric tests of different algorithms in higher dimensions.

Dimension	Algorithm	Rank	Average Friedman Value	+/=/−
*D* = 50	GWO	6	5.2225	22/0/1
	HHO	4	4.9471	19/3/1
	ZOA	3	3.8826	20/3/0
	SMA	2	2.8232	19/3/1
	SCA	9	8.1261	22/0/1
	AROA	8	7.1536	23/0/0
	RIME	7	6.0232	22/0/1
	WOA	5	5.1471	22/0/1
	LSEWOA	**1**	**1.6746**	−
*D* = 100	GWO	6	5.3551	22/0/1
	HHO	4	4.8101	19/3/1
	ZOA	3	3.8442	20/3/0
	SMA	2	2.7645	19/3/1
	SCA	9	8.3072	22/0/1
	AROA	8	6.9710	23/0/0
	RIME	7	6.2884	22/0/1
	WOA	5	5.1043	22/1/0
	LSEWOA	**1**	**1.5551**	−

**Table 15 sensors-25-02054-t015:** Average fitness and standard deviation of each algorithm across the seven engineering design problems.

Challenges	Metrics	GWO	HHO	ZOA	SMA	SCA	AROA	RIME	WOA	LSEWOA
Three-bar Truss	Ave	259.805063	259.815011	259.805048	263.072647	259.820148	259.832780	259.806407	259.863959	259.805047
	Std	0.000015	0.014659	0.000001	2.666551	0.012243	0.083901	0.001547	0.081753	0.000000
Tension/Compression Spring	Ave	0.121526	0.121522	0.121523	0.121522	0.121740	0.124473	0.122241	0.121921	0.121522
	Std	0.000005	0.000000	0.000001	0.000000	0.000231	0.009336	0.003592	0.001725	0.000000
Speed Reducer	Ave	2638.848210	2638.824969	2638.820667	2638.819863	2647.800460	2640.757902	2638.866459	2638.820024	2638.819842
	Std	0.025308	0.023320	0.000996	0.000062	6.014099	2.752896	0.101059	0.000388	0.000020
Cantilever Beam	Ave	13.360394	13.390888	13.360290	13.360645	13.963792	20.620153	13.584452	15.444253	13.360259
	Std	0.000108	0.021638	0.000050	0.000308	0.212685	4.997151	0.179248	1.779201	0.000000
I-beam	Ave	6.702705	6.702689	6.702962	6.703047	6.664008	5.782476	6.457600	6.365987	6.703048
	Std	0.000320	0.000531	0.000105	0.000001	0.035199	0.995907	0.289582	0.322097	0.000000
Piston Lever	Ave	12.179036	274.057852	2.953538	34.340337	1.219121	281.004695	52.942343	28.643545	1.057195
	Std	42.317243	238.155716	7.405989	67.704274	0.071245	202.116656	132.158718	79.421364	0.000099
Multi-disc Clutch Brake	Ave	0.235302	0.235243	0.235258	0.235243	0.237668	0.236633	0.235452	0.235244	0.235242
	Std	0.000082	0.000001	0.000015	0.000001	0.002042	0.002617	0.000278	0.000009	0.000000
Gas Transmission System	Ave	1224745.959830	1224745.937224	1224745.938295	1224745.937223	1224901.598433	1226318.258250	1224745.952955	1224745.937227	1224745.937222
	Std	0.017907	0.000005	0.001656	0.000002	118.092819	3538.385964	0.022237	0.000007	0.000000
Industrial Refrigeration System	Ave	642.809538	897.655989	13.111876	642.333923	9.729020	23312.391077	8.430037	868.334558	8.249197
	Std	3477.950844	3927.456866	5.330660	3475.959550	1.070188	24530.767002	2.012544	4131.221271	0.496502

## Data Availability

No dataset is used in this reasearch.

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
