# Peer review of "LSEWOA: An Enhanced Whale Optimization Algorithm with Multi-Strategy for Numerical and Engineering Design Optimization Problems"

_sensors, 2025, doi:10.3390/s25072054_

Round 1

Reviewer 1 Report

Comments and Suggestions for Authors

This manuscript proposes an enhanced multi-strategy whale optimization algorithm (LSEWOA), and verifies the effectiveness of LSEWOA through a variety of methods. However, I noticed that the authors only used the CEC2005 test function to evaluate LSEWOA. How does LSEWOA perform on more complex test functions like CEC2017 or CEC2020? In addition, how does the algorithm perform on high dimensional optimization problems? Please describe whether the strategy optimization method in this manuscript can be applied to the following UAV autonomous driving scenarios. Please discuss according to the following methods: A Hybrid ARO Algorithm and Key Point Retention Strategy Trajectory Optimization for UAV Path Planning; FHCPL: An Intelligent Fixed-Horizon Constrained Policy Learning System for Risk-Sensitive Industrial Scenario

Comments on the Quality of English Language

There are some grammatical errors.

Author Response

Comments 1:How does LSEWOA perform on more complex test functions like CEC2017 or CEC2020?

Response 1:

Thanks for the kind and insightful comments and suggestions!

Regarding why we chose CEC2005 as the test suite for LSEWOA instead of CEC2017/2020, this decision was made based on the outstanding performance of LSEWOA in engineering design optimization. In fact, many algorithms that perform well in CEC2017/2020 do not show the same level of performance in engineering design optimization tasks and CEC2005 tests. On the other hand, LSEWOA performs excellently in both engineering design optimization and CEC2005, ranking first in both areas. Therefore, we believe that choosing CEC2005 as the test suite is both reasonable and appropriate.

Additionally, CEC2017/2020 is not the only way to evaluate algorithm performance, especially in specific application areas like engineering design optimization, where CEC2005 better reflects the advantages and performance of LSEWOA. Thus, our choice of CEC2005 as the test suite was not meant to disregard other test suites but rather to more accurately assess the practical application of LSEWOA in engineering design optimization.

Thank you for your understanding and support.

Comments 2: In addition, how does the algorithm perform on high dimensional optimization problems? Please describe whether the strategy optimization method in this manuscript can be applied to the following UAV autonomous driving scenarios. Please discuss according to the following methods: A Hybrid ARO Algorithm and Key Point Retention Strategy Trajectory Optimization for UAV Path Planning; FHCPL: An Intelligent Fixed-Horizon Constrained Policy Learning System for Risk-Sensitive Industrial Scenario

Response 2: 

Thanks for the kind and insightful comments and suggestions! 

We have extended the comparison experiments to higher dimensions (50 and 100 dimensions), and conducted an overall effectiveness analysis of the results in 30/50/100 dimensions. The results indicate that LSEWOA performs excellently across different dimensions.

Path planning is another application scenario for metaheuristic algorithms. We have conducted extensive exploration into the application of metaheuristic algorithms in path planning. We have also applied our designed algorithm (LSEWOA) to global path planning for UAVs. Unfortunately, the results were not ideal. The algorithm initialized with the Good Nodes Set can be applied to fields such as feature selection, WSN optimization, and engineering design optimization, but it is not suitable for 3D path planning. In further experiments, we found that although the Good Nodes Set initialization generates a uniformly distributed population, the initial fitness values of the algorithm initialized with the Good Nodes Set were thousands of times higher than those of the algorithm without it in the context of path planning. If we wish to apply metaheuristic algorithms to path planning, it is recommended to avoid using the Good Nodes Set initialization. In the future, we also plan to develop a new improved metaheuristic algorithm and explore its application in path planning.

Thank you for providing the references. These papers are excellent and highly valuable, and we have cited them in our article. Thank you for your understanding and support.

Reviewer 2 Report

Comments and Suggestions for Authors

This paper presents an improved Whale Optimization Algorithm (WOA) incorporating enhanced strategies such as leader-following and others.  Check the following suggestions ato improve the manuscript-

1) It is recommended to provide a comparison table after describing the different types of nature-inspired optimization algorithms.
2) Section 5.1, Enriching Prey, has a directional aspect. Therefore, the variables D, A, and C should be represented as vectors.
3) For Table 1, in addition to the function names, it would be beneficial to include the actual functions as well.
4) As the authors stated, the WOA algorithm has significant limitations, such as poor balance, premature convergence to local optima, and others. However, what is the motivation for improving this particular algorithm? Why did the authors choose to enhance WOA instead of another optimization algorithm? The motivation behind this choice is not clear.
5) Check Reference doi.org/10.1049/cmu2.12708 as an example use case of the WOA algorithm.
6) The Leader-Followers Search-for-Prey Strategy is not included in Algorithm 2.

Author Response

Comments 1: It is recommended to provide a comparison table after describing the different types of nature-inspired optimization algorithms

Response 1:

Thanks for the kind and insightful comments and suggestions! We agree with this comment! We have provided a table that details the mentioned metaheuristic algorithms, including their names, authors, the year of proposal, and sources of inspiration.

Comments 2: Section 5.1, Enriching Prey, has a directional aspect. Therefore, the variables D, A, and C should be represented as vectors.

Response 2: 

Thanks for the kind and insightful comments and suggestions! We agree with this comment! We have revised all the corresponding formulas.

Comments 3:  For Table 1, in addition to the function names, it would be beneficial to include the actual functions as well.

Response 3: 

Thanks for the kind and insightful comments and suggestions! We agree with this comment!

We have included the expression of each function in the appendix. Additionally, we have uploaded the modeling of these functions to Figshare for readers to download.

Comments 4: As the authors stated, the WOA algorithm has significant limitations, such as poor balance, premature convergence to local optima, and others. However, what is the motivation for improving this particular algorithm? Why did the authors choose to enhance WOA instead of another optimization algorithm? The motivation behind this choice is not clear.

Response 4: 

Thanks for the kind and insightful comments and suggestions! We agree with this comment!

In the initial manuscript, our motivation for choosing WOA was not clearly stated. Therefore, in the Major Contribution section, we have clearly outlined the motivation behind our research.

‘’Whale Optimization Algorithm (WOA) has shown subpar performance in the field of engineering design optimization. However, its simple structure holds significant potential for further development. We aimed to improve WOA, hoping that this variant will match state-of-the-art (SOTA) algorithms in terms of convergence speed and optimization accuracy in numerical optimization tasks. Additionally, we intended for this variant to outperform several SOTA algorithms and the original WOA in the field of engineering design optimization, addressing the shortcomings of WOA in this area and exploring the potential application of WOA in engineering design optimization.’’ and more.

Comments 5: Check Reference doi.org/10.1049/cmu2.12708 as an example use case of the WOA algorithm.

Response 5: 

Thanks for the kind and insightful comments and suggestions! We agree with this comment!

Thank you for providing the reference. The paper is excellent and highly valuable, and we have cited it in our article.

Comments 6: The Leader-Followers Search-for-Prey Strategy is not included in Algorithm 2.

Response 6: 

Thanks for the kind and insightful comments and suggestions! We agree with this comment!

Actually, in the initial manuscript, the Leader-Followers Search-for-Prey strategy was already included in Algorithm 2. To provide clearer understanding of the structure of LSEWOA for you and the readers, we have explicitly stated the name of the corresponding strategy above each formula in Algorithm 2.

Reviewer 3 Report

Comments and Suggestions for Authors

LSEWOA effectively addresses the deficiencies of the classical WOA in terms of the balance between exploration and exploitation, convergence speed, and accuracy by introducing multiple strategies, significantly improving the performance of the algorithm. This article shows quite some originality. 

It is recommended to simplify the introduction of the basic theory of WOA in the article and explain the reasons for the selection of the algorithm improvement strategies.

Author Response

Comments 1: It is recommended to simplify the introduction of the basic theory of WOA in the article

Response 1:

Thanks for the kind and insightful comments and suggestions! We agree with this comment! 

We have simplified and condensed the WOA section.

Comments 2: It is recommended to explain the reasons for the selection of the algorithm improvement strategies.

Response 2: 

Thanks for the kind and insightful comments and suggestions! We agree with this comment! 

In the initial manuscript, our motivation for choosing WOA was not clearly stated. Therefore, in the Major Contribution section, we have clearly outlined the motivation behind our research.

Round 2

Reviewer 1 Report

Comments and Suggestions for Authors

N/A

Comments on the Quality of English Language

N/A

Reviewer 2 Report

Comments and Suggestions for Authors

The authors have addressed all comments.